# FANTASYWORLD: GEOMETRY-CONSISTENT WORLD MODELING VIA UNIFIED VIDEO AND 3D PREDICTION

**Yixiang Dai[1][*][§], Fan Jiang[1][*][†][‡], Chiyu Wang[1][*], Mu Xu[1], Yonggang Qi[2][‡]**

[1]AMAP, Alibaba Group
[2]Beijing University of Posts and Telecommunications
daiyixiang666@gmail
{frank.jf, wangchiyu.wcy, xumu.xm}@alibaba-inc.com
qiyg@bupt.edu.cn

## ABSTRACT

High-quality 3D world models are pivotal for embodied intelligence and Artificial General Intelligence (AGI), underpinning applications such as AR/VR content creation and robotic navigation. Despite the established strong imaginative priors, current video foundation models lack explicit 3D grounding capabilities, thus being limited in both spatial consistency and their utility for downstream 3D reasoning tasks. In this work, we present FANTASYWORLD, a geometry-enhanced framework that augments frozen video foundation models with a trainable geometric branch, enabling joint modeling of video latents and an implicit 3D field in a single forward pass. Our approach introduces cross-branch supervision, where geometry cues guide video generation and video priors regularize 3D prediction, thus yielding consistent and generalizable 3D-aware video representations. Notably, the resulting latents from the geometric branch can potentially serve as versatile representations for downstream 3D tasks such as novel view synthesis and navigation, without requiring per-scene optimization or fine-tuning. Extensive experiments show that FANTASYWORLD effectively bridges video imagination and 3D perception, outperforming recent geometry-consistent baselines in multi-view coherence and style consistency. Ablation studies further confirm that these gains stem from the unified backbone and cross-branch information exchange. Project page: https://fantasy-amap.github.io/fantasy-world/.

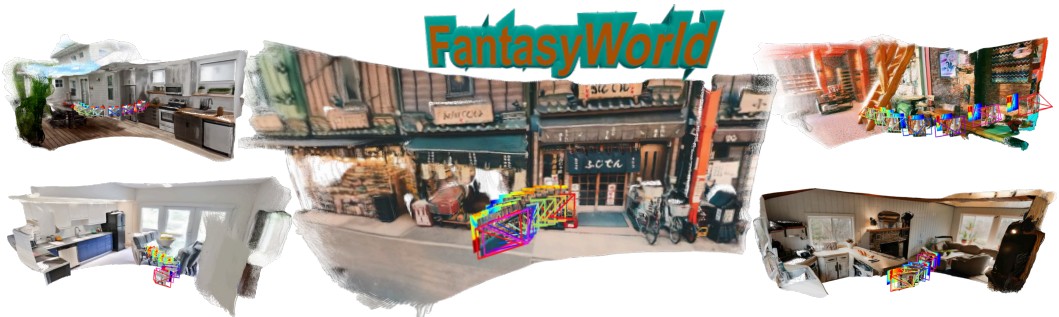

Figure 1: FANTASYWORLD overview. Given multimodal inputs (image, text, and camera trajectory), the model generates photorealistic videos along the specified views while constructing an implicit 3D representation for consistent geometry.

---

[*]Equal contribution.

[†]Project leader.

[‡]Corresponding author.

[§]Work done during internship at AMAP, Alibaba Group.

# 1 INTRODUCTION

Building world models has long been viewed as a key step toward Artificial General Intelligence (AGI). By modeling environments, objects, causal relations, and temporal dynamics, world models enable agents to predict, plan, and generate, endowing them with human-like understanding and creative abilities. At the core of this vision lies the construction of high-quality, diverse 3D environments, which support a wide spectrum of applications spanning AR/VR content creation, robotic navigation, and embodied AI at large. Plenty of approaches have been explored towards the goal of generating coherent 3D scenes, involving leveraging 2D generative priors to enable 3D generation (Chen et al., 2023; Bahmani et al., 2024a) via SDS (Poole et al., 2022), building entire 3D scenes by multi-view images with NeRF (Mildenhall et al., 2021) and 3DGS (Kerbl et al., 2023).

Recently, camera-guided video diffusion models have gained popularity, where image frames implicitly encode the scene's 3D structure with multiple images from different viewpoints, as videos are natural 2D projections of dynamic 3D worlds (Huang et al., 2025a; Cao et al., 2025; He et al., 2024; Bahmani et al., 2025; 2024b). Such methods harness the spatiotemporal capacity of video diffusion to improve 3D consistency and multi-view modeling, such as ReconX (Liu et al., 2024a), Gen3C (Ren et al., 2025), DimensionX (Sun et al., 2024), and ViewCrafter (Yu et al., 2024b), to name a few. These models excel at producing geometrically plausible local shapes and mitigating view inconsistency through temporal priors. Beyond perspective video generation, another related line of work focuses on generating panoramic 360 degree content, such as images and videos, to create immersive environments for exploration. Panoramic viewpoints provide a much larger field of view than standard perspective cameras, but they also introduce strong geometric distortions and suffer from limited training data (Wallingford et al., 2024; Lu et al., 2024; Wang et al., 2024a; Li et al., 2024; Zhou et al., 2025; Xing et al., 2025; Team et al., 2025b; Liu et al., 2025a).

However, despite being trained on vast Internet video corpora and exhibiting strong imaginative priors, generative video models lack explicit 3D supervision, making it difficult to preserve real-world structure and spatial consistency. This reveals a central challenge in world model construction: *how to inject reliable geometric grounding into video generation without sacrificing creative capacity*.

Lately, the emergence of 3D foundation models, such as DUSt3R (Wang et al., 2024b), MASt3R (Duisterhof et al., 2025), Fast3R (Yang et al., 2025a), and VGGT (Wang et al., 2025a), demonstrates that robust geometry can be predicted in a single forward pass without additional 3D reconstruction, providing a scalable solution to geometry-consistent generation. Consequently, several works attempt to couple 2D video generation with 3D reasoning. Voyager (Huang et al., 2025a) trains an end-to-end model that jointly predicts RGB and depth while maintaining world consistency through a cache and geometry-injected frames, and then reconstructs explorable 3D scenes. Geometry Forcing (Wu et al., 2025) further marries video diffusion with an explicit 3D representation to strengthen geometric consistency during training. Matrix-3D (Yang et al., 2025c) shows two practical routes from omni-directional video to a navigable world, including an optimization path and a large panorama reconstruction model that directly infers 3D Gaussians from video latents. WonderWorld (Yu et al., 2025a) focuses on interactive single-image scene authoring with layered Gaussian surfels and guided depth diffusion for fast, connected scene creation. Vidu4D (Wang et al., 2024d) shows that even a single generated video can support high-quality 4D reconstruction when paired with dynamic Gaussian surface elements. Complementary to these trends in content creation, GaussianWorld (Zuo et al., 2025) frames perception as streaming 4D occupancy forecasting, underscoring the value of a 3D world representation that can generalize across embodied tasks.

Despite significant progress in video-to-3D modeling, several limitations remain. First, most video generative models operate purely within the video domain, yielding features that cannot directly support 3D reasoning. When explicit 3D reconstruction is desired, they often resort to additional scene-specific optimization with NeRF (Mildenhall et al., 2021) or 3DGS (Kerbl et al., 2023), which introduces computation overhead. Second, video imagination and 3D perception remain weakly coupled at inference, preventing mutual reinforcement, which effectiveness is evidenced by recent works in 3D scene understanding (Huang et al., 2025b). For instance, although Voyager (Huang et al., 2025a) predicts RGB and depth jointly to maintain world consistency, and WorldExplorer (Schneider et al., 2025) leverages video-based imagination for scene exploration, in both cases the two processes operate largely independently, highlighting the persistent limitation of weak coupling between video generation and 3D perception. Third, many approaches, such as Geometry Forcing

(Wu et al., 2025), integrate 3D priors by fine-tuning video foundation models (VFMs) while keeping a 3D model like VGGT frozen, which incurs substantial computational cost and risks compromising the VFM's general generative capacity.

To address these limitations, we introduce FANTASYWORLD, a geometry-enhanced framework that efficiently produces reusable 3D-consistent features by augmenting frozen VFMs with an additional trainable branch for geometric inference, tightly coupling video imagination and 3D perception without expensive per-scene optimization or fine-tuning. Instead of predicting depth or point clouds from RGB images, we directly infer camera parameters and 3D signals from video latents. This is inspired by VGGT but achieves tighter integration between generative and geometric modeling. Concretely, we split the backbone of video foundation models (i.e., Wan2.1 in our case) into Preconditioning Blocks (PCB) that inject video priors and stabilize latents, and Integrated Reconstruction and Generation Blocks (IRG) that fuse spatiotemporal tokens with a geometry co-encoder to predict a geometry-aware implicit 3D field. As a result, our model generates camera-conditioned video features alongside an explicit 3D representation in a single forward pass, without relying on additional 3D reconstruction (e.g., NeRF or 3DGS) or iterative memory refinement as in Voyager ((Huang et al., 2025a)).

To this end, our contributions are as follows: (i) Unified video-3D modeling: We propose FANTASYWORLD, a geometry-enhanced framework that jointly predicts video latents and an implicit 3D field through a single backbone, preserving imaginative priors while exposing explicit geometry. (ii) 2D/3D cross-branch supervision: We introduce constraints that let geometry supervise video features and video priors regularize 3D prediction, ensuring 3D-consistent frames inference. (iii) Potential for generalizable 3D features: We expect that the resulting video-3D representations serve as versatile features for downstream tasks, such as novel view synthesis and navigation, without per-task adaptation, which has been evidenced by recent works, such as AnySplat (Jiang et al., 2025).

## 2 RELATED WORK

### 2.1 FEED-FORWARD RECONSTRUCTION

Feed-forward reconstruction methods have achieved promising results in recovering 3D properties of a scene from a set of images in a single pass (Wang et al., 2024c; 2025b; Wang & Agapito, 2024; Duisterhof et al., 2025; Tang et al., 2025; Yang et al., 2025b; Zhang et al., 2024; 2025b; Wang et al., 2025a;c). DUSt3R (Wang et al., 2024c) and MASt3R (Duisterhof et al., 2025) directly estimate point clouds from images, which is suitable for challenging scenarios such as low-texture regions, but can only input two images at a time. Fast3R (Yang et al., 2025b) can process thousands of frames at once, surpassing the aforementioned restrictions. VGGT (Wang et al., 2025a) introduces a more generalized framework capable of supporting input from one frame to multiple frames and producingg multiple 3D attributes. Furthermore, its feature backbone serves as a versatile feature extractor for various downstream tasks. Recently, many methods have integrated diffusion knowledge into reconstruction techniques, thereby enriching the 3D modeling process with the ability to imagine occluded or unobserved areas (Fu et al., 2024; Hu et al., 2025; Ke et al., 2024; Lu et al., 2025; Yang et al., 2024; Zhu et al., 2024; 2023; Liang et al., 2025). In this work, we adopt an architecture similar to VGGT as our 3D implicit feature extractor. Specifically, we extract implicit 3D features from WanDiT block and integrate them within a multi-task learning paradigm.

### 2.2 GEOMETRY-AWARE VIDEO GENERATION

The consistency of generated videos with the real physical world is a crucial challenge in simulation. Many previous works have attempted to incorporate geometric constraints during the video generation process, primarily categorized into explicit guidance and implicit guidance. For explicit guidance, many prior works explicitly incorporate 3D signals (such as point clouds, mesh, etc.) obtained from the first frame into the model, thereby demonstrably improving generation consistency (Yu et al., 2024a; Huang et al., 2025a; Yang et al., 2025d; Cao et al., 2025). These methods often suffer from insufficient point cloud accuracy and limited scope when subjected to significant viewpoint changes. For implicit guidance, many prior works investigate methods to enable models to perceive 3D structural information within the diffusion process (Zhang et al., 2025a; Team et al.,

2025a; Wu et al., 2025). Geometry-Forcing (Wu et al., 2025) aligns the model's intermediate representations with the features of a pre-trained geometric foundational model, risks undermining the creativity of large-scale video generation models trained on massive video data.

### 2.3 JOINT 3D AND VIDEO GENERATION

Integrating both video and 3D structural information enables the generation of comprehensive 3D scenes for applications in embodied AI and simultaneously facilitates a better representation of the underlying physical world. AETHER (Team et al., 2025a) unifies RGB–depth modeling and couples reconstruction with video generation. DeepVerse (Chen et al., 2025a) achieves interactive joint generation via an autoregressive paradigm and control via text-specified control signals. Voyager (Huang et al., 2025a) incorporates point cloud projection within its joint generation framework to achieve camera control.

In this paper, we introduce an implicit 3D feature encoding branch designed to decode various 3D attributes. Through mutual enhancement achieved during the diffusion process, it simultaneously yields video features and 3D structural features.

## 3 METHODOLOGY

### 3.1 OVERVIEW

FANTASYWORLD is a unified feed-forward model for joint video and 3D scene generation. Given a reference image, an optional text prompt, and a target camera trajectory, the model produces a video aligned with the specified views while simultaneously constructing an implicit 3D representation. Inputs are encoded by pretrained backbones: CLIP (Radford et al., 2021) for images, umT5 (Chung et al., 2023) for text, and a learned camera encoder, following Wan's Plücker-ray design (Wan et al., 2025), for camera poses. These signals jointly condition both the video and geometry branches during training and inference.

As shown in Fig. 2 (a), the front end employs *Preconditioning Blocks (PCBs)* that reuse the frozen WanDiT denoiser to supply partially denoised latents, ensuring the geometry pathway operates on meaningful features rather than pure noise. The backbone then consists of stacked *Integrated Reconstruction and Generation (IRG) Blocks*, which iteratively refine video latents and geometry features under multimodal conditioning. Each IRG block contains an asymmetric dual-branch structure (Fig. 2 (b)): an *Imagination Prior Branch* for appearance synthesis and a *Geometry-Consistent Branch* for explicit 3D reasoning, coupled through lightweight adapters and cross attention.

The outputs are twofold. The imagination branch generates geometry-consistent video frames along the trajectory, while the geometry branch produces task-agnostic 3D features decoded by DPT heads into depth maps, point maps, and camera poses. This unified design supports downstream tasks such as novel view synthesis, pose estimation, and depth prediction, all without per-scene optimization.

In summary, FANTASYWORLD bridges generative video priors with structured geometric reasoning in a single forward pass, producing outputs that are both photorealistic and geometrically consistent.

### 3.2 PRECONDITIONING BLOCKS

Denoising diffusion models progressively remove noise across timesteps, revealing structure and details in the signal. Recent theory (Han et al., 2025) shows that the denoising objective balances learning signal and noise, enabling structural information to emerge gradually—consistent with the empirical observation that features become more informative as denoising unfolds. We further observe a similar effect along network depth: even at a fixed timestep, deeper WanDiT layers produce clearer spatial structure (Fig. 3), suggesting that denoising progresses across both time and depth.

Motivated by this, we introduce *Preconditioning Blocks (PCBs)* at the front end of our framework. We reuse the first 16 frozen layers of Wan2.1, following its camera conditioning design, to partially denoise video latents. This ensures that inputs to the geometry branch contain geometry-relevant cues rather than pure noise, reducing gradient variance and avoiding early training dominated by

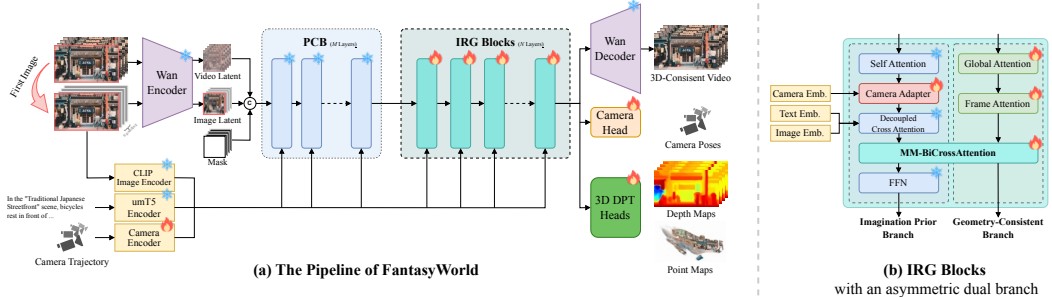

**(a) The Pipeline of FantasyWorld**

**(b) IRG Blocks**
with an asymmetric dual branch

Figure 2: Overview of FANTASYWORLD. Inputs (image, text, camera) are processed by PCBs and stacked IRG blocks, where an asymmetric dual-branch design couples video synthesis with 3D reasoning. The model outputs geometry-consistent video frames and task-agnostic 3D features.

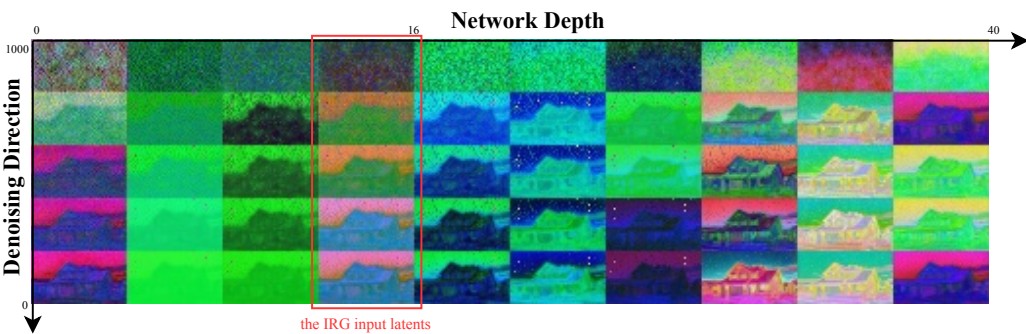

Figure 3: PCA over timestep–block pairs: rows vary timesteps top → bottom, columns vary blocks left → right; the red rectangle marks the IRG input latents.

high-noise latents. PCBs thus bridge noisy initialization and geometry-aware processing, providing stable supervision from the start and allowing the geometry branch to focus on refining structure.

### 3.3 INTEGRATED RECONSTRUCTION AND GENERATION BLOCKS

At the core of FANTASYWORLD are the *Integrated Reconstruction and Generation (IRG) Blocks*, stacked as the fundamental units of the model. Each IRG block adopts an asymmetric dual-branch design (Fig. 2 (b)): the *Imagination Prior Branch* reuses the pretrained Wan2.1 backbone to propagate appearance-rich spatiotemporal features, while the *Geometry-Consistent Branch* projects them into a geometry-aligned latent space. Unlike VGGT, which relied on DINO features, we bridge the geometry pathway to Wan latents, ensuring geometry is inferred in the same domain as video synthesis and avoiding feature mismatch.

IRG blocks unify the two branches through bidirectional cross-attention (MM-BiCrossAttn (Liu et al., 2024b; 2025b)). Here, geometric cues regularize video features for multi-view coherence, while video priors provide imaginative signals that complete occluded regions and refine geometry. As cooperative units, stacked IRG blocks progressively enhance both video latents and geometry features, forming the core mechanism where imagination and structure converge.

The geometry branch outputs an implicit representation decoded by a custom 3D DPT head. This head performs temporal decoding aligned with the WanVAE video frames. Motivated by the depth-wise emergence of structure observed in Sec. 3.2, the head inverts the conventional reassemble strategy: instead of drawing fine features from early encoder layers dominated by noise, it sources them from late diffusion blocks where semantics are stronger and denoising is mature. Anchoring predictions in these stable features improves depth accuracy, stabilizes pose estimation, and enhances the consistency of the implicit 3D field.

### 3.4 BRIDGE TO UNIFY TRAINING

**Two-stage framework.** Training in FANTASYWORLD follows two stages: first *bridging* the geometry branch to the Wan feature space, then *unifying* geometry and video through bidirectional cross-attention. Stage 1 freezes the Wan2.1 backbone and adapts the geometry branch to consume hidden features; Stage 2 introduces cross-branch adapters for joint optimization, while keeping the backbone frozen. Please refer to A.2, A.3 A.4 for more implementation and training details.

**Stage 1: Latent Bridging.** We select hidden features from block 16 of Wan2.1 and feed them to the geometry branch through a lightweight transformer adapter that maps to a geometry-aligned latent space. This latent is encoded and decoded into camera, depth, and point map predictions, while only the geometry branch is trained. Supervision is applied via:

$$\mathcal{L}_{\text{geo}} = \alpha \, \mathcal{L}_{\text{depth}} + \beta \, \mathcal{L}_{\text{pmap}} + \gamma \, \mathcal{L}_{\text{camera}}$$

where $\mathcal{L}_{\text{depth}}$ follows Video Depth Anything (Chen et al., 2025b), $\mathcal{L}_{\text{pmap}}$ follows VGGT (Wang et al., 2025a), and $\mathcal{L}_{\text{camera}}$ is a Huber penalty. This stage ensures the geometry branch operates stably on Wan latents instead of raw noise.

**Stage 2: Unified Co-Optimization.** From block 16 onward, we insert one bidirectional cross-attention adapter after each of 24 transformer blocks, aligned with the geometry branch. For video tokens $X_v$ and geometry tokens $X_g$, with projections $(Q_v, K_v, V_v)$ and $(Q_g, K_g, V_g)$, attention is:

$$A = \text{softmax}\left( \frac{Q_v K_g^\top}{\sqrt{d_k}} \right)$$

and updates are:

$$X_v^+ = X_v + \gamma_v A V_g, \qquad X_g^+ = X_g + \gamma_g A^\top V_v$$

with learnable gates $\gamma_v, \gamma_g$. Geometry-to-video updates enforce 3D consistency, while video-to-geometry updates inject generative priors. Camera parameters are embedded with a pose encoder (Wan et al., 2025); we modify the pose adapter to predict only the shift $\beta_i$, injected additively:

$$f_i = f_{i-1} + \beta_i,$$

applied to the first 24 of 40 blocks.

**Training objective.** The final objective is:

$$\mathcal{L}_{\text{total}} = \mathbb{E}_{z_0, \epsilon, t, c} \left[ \| \epsilon_\theta(z_t, t, c) - \epsilon \|_2^2 \right] + \lambda \mathcal{L}_{\text{geo}}$$

combining the standard diffusion loss with geometry supervision which aggregates depth, point map, and camera supervision. The weight $\lambda$ balances video generation and geometry learning, enforcing multi-view coherence and enabling cross-branch co-adaptation.

## 4 EXPERIMENT

We conduct comprehensive experiments to evaluate FANTASYWORLD across large-scale benchmarks and diverse scenarios. Sec. 4.1 outlines datasets, training protocols, and evaluation settings. Sec. 4.2 assesses world generation, including comparisons with state-of-the-art baselines, analysis under varying camera motions, and ablations on the geometry branch. Sec. 4.3 examines geometric fidelity through quantitative reconstruction metrics and qualitative visualizations, further validating the role of explicit geometry modeling.

### 4.1 IMPLEMENTATION DETAILS

**Datasets.** Our training corpus uses approximately 180k video clips from a diverse mix of real-world and simulated sources, with geometric supervision obtained via multiple strategies. For RealEstate10K (Zhou et al., 2018) and ACID (Liu et al., 2021), we generate multi view consistent depth maps using a reconstruction based pipeline. For DL3DV (Ling et al., 2024), WildRGB (Xia et al., 2024), ScanNet (Dai et al., 2017), and the simulated TartanAir (Wang et al., 2020), we instead apply Cut3R (Wang et al., 2025b) to extract geometric labels.

**Training Details.** We train the model with AdamW and a learning rate of $10^{-5}$ in two stages. In Stage 1 (latent bridging), the geometry branch is trained for 20,000 steps with a global batch size of 64, while the video backbone remains frozen, adapting the geometry branch to its feature space. In Stage 2 (unified co-optimization), training uses 81 frame clips at $592 \times 336$ and $336 \times 592$. Only the lightweight interaction modules (bidirectional cross attention and camera control adapter) are fine tuned for 10,000 steps with a global batch size of 112, with all core backbones frozen. Stage 1 runs on 64 H20 GPUs for 36 hours, and Stage 2 on 112 H20 GPUs for 144 hours.

**Evaluation.** We conduct evaluation on 1,000 samples drawn from the photorealistic subset of the WorldScore static benchmark (Duan et al., 2025). The static split of WorldScore is divided into photorealistic and stylized subsets. Since our objective is to model real-world environments for embodied intelligence, we focus on photorealistic video synthesis. In contrast, evaluating 3D consistency in stylized videos is ill-defined and less representative of performance in realistic scenarios, and is therefore excluded from our study. We assess our model along two complementary dimensions:

- **World Generation**: For a holistic evaluation of controllable world generation, we report performance on the WorldScore benchmark, which measures performance across multiple axes, including camera and object control, content alignment, 3D consistency, and perceptual quality.
- **Geometric Fidelity**: To quantify geometric fidelity, we reconstruct each scene with 3DGS, and report PSNR, SSIM, and LPIPS.

Together, these metrics capture both the controllability of video-based world generation and the structural accuracy of 3D geometry. For more evaluation details, please refer to A.5.

## 4.2 WORLD GENERATION

We evaluate the video generation capabilities of FANTASYWORLD by comparing it against WonderWorld (Yu et al., 2025a), Uni3C (Cao et al., 2025), Voyager (Huang et al., 2025a), and AETHER (Team et al., 2025a), which represent the most relevant state-of-the-art baselines for 3D world generation.

**Quantitative Comparison.** We evaluate under two camera-motion settings. The *Small* configuration follows the WorldScore static benchmark, using 1,000 photorealistic samples to ensure consistency with prior work. We additionally introduce a *Large* setting with 100 curated cases featuring wide orbital or panning trajectories (up to 90°), to test robustness under challenging viewpoints. As shown in Table 1, FANTASYWORLD achieves the highest scores on all consistency-related metrics (*3D Consist.*, *Photo Consist.*, and *Style Consist.*). Moreover, it yields the lowest standard deviation across samples, suggesting that our method is not only more accurate on average but also more stable across diverse scenes. In the Large setting, the gains are particularly pronounced, demonstrating strong geometric stability even under substantial viewpoint shifts. Compared to baselines that prioritize instruction following and camera manipulation (Yu et al., 2025a), our framework focuses instead on embedding geometric awareness into video features. Although this design does not directly optimize camera or content-alignment benchmarks, it enables stable and reusable 3D representations (our main focus), and can be extended with post-hoc explicit reconstruction to recover strong camera controllability when needed (see Appendix A.6).

**Qualitative Analysis.** Under large camera motion, distinct failures emerge: WonderWorld shows tearing and holes indicative of missing geometry; in Uni3C and Voyager, first-frame point-cloud priors quickly fall out of view, causing style drift in Uni3C and multi-view misalignment in Voyager; AETHER, despite generating RGB-D point clouds, often produces incoherent, low-detail content (see Fig. 4). In contrast, FANTASYWORLD predicts an implicit 3D representation that evolves with the video, ensuring stable geometry and appearance across time and yielding coherent, 3D-consistent results without the failures of static priors.

**Ablation on the Geometry Branch.** To assess the role of explicit geometry modeling, we compare our full model with a variant where the geometry branch and bidirectional cross-attention are removed. As shown in Table 1, removing the geometry branch leads to declines in *Photo Consist.* and *Style Consist.*, and an especially severe drop in *3D Consist.*, underscoring its critical role in ensuring multi-view coherence and high-fidelity generation.

Table 1: WorldScore with *Small* vs. *Large* camera motion.

| Method | Motion | 3D Consist. | Photo Consist. | Style Consist. | Camera Ctrl. | Object Ctrl. | Content Align. | Subjective Qual. |
|---|---|---|---|---|---|---|---|---|
| WonderWorld | | $82.85_{\pm19.69}$ | $67.86_{\pm23.56}$ | $55.79_{\pm34.89}$ | **92.32** | 47.63 | **79.09** | **69.03** |
| AETHER | | $79.84_{\pm14.68}$ | $58.68_{\pm38.59}$ | $72.09_{\pm32.62}$ | 57.44 | 52.26 | 28.06 | 41.11 |
| Uni3C | Small | $78.59_{\pm21.08}$ | $85.48_{\pm20.98}$ | $88.32_{\pm18.47}$ | 62.94 | 45.83 | 47.40 | 57.00 |
| Voyager | | $56.00_{\pm26.32}$ | $80.68_{\pm16.32}$ | $72.89_{\pm29.78}$ | 45.92 | **57.69** | 48.36 | 44.74 |
| **Ours w/o 3D** | | $79.77_{\pm16.06}$ | $83.86_{\pm8.73}$ | $92.54_{\pm12.90}$ | 57.94 | 37.33 | 43.31 | 55.85 |
| **Ours w/ 3D** | | $\mathbf{83.31}_{\pm14.24}$ | $\mathbf{86.11}_{\pm7.97}$ | $\mathbf{94.22}_{\pm9.11}$ | 57.05 | 34.46 | 38.45 | 57.40 |
| WonderWorld | | $63.70_{\pm24.37}$ | $3.22_{\pm8.47}$ | $35.95_{\pm33.47}$ | **96.28** | 38.61 | **97.10** | **72.46** |
| AETHER | | $63.97_{\pm17.39}$ | $33.11_{\pm23.99}$ | $61.99_{\pm32.24}$ | 4.43 | 34.78 | 33.69 | 35.09 |
| Uni3C | Large | $73.95_{\pm17.55}$ | $46.78_{\pm32.64}$ | $71.43_{\pm29.38}$ | 8.69 | 34.28 | 77.88 | 51.12 |
| Voyager | | $13.82_{\pm19.96}$ | $9.52_{\pm17.17}$ | $61.34_{\pm35.29}$ | 0.00 | **49.23** | 64.10 | 39.21 |
| **Ours w/o 3D** | | $72.06_{\pm20.14}$ | $56.98_{\pm23.60}$ | $81.59_{\pm22.23}$ | 9.32 | 34.44 | 75.85 | 46.96 |
| **Ours w/ 3D** | | $\mathbf{74.83}_{\pm16.31}$ | $\mathbf{60.61}_{\pm21.39}$ | $\mathbf{82.02}_{\pm19.56}$ | 11.24 | 31.96 | 77.20 | 50.46 |

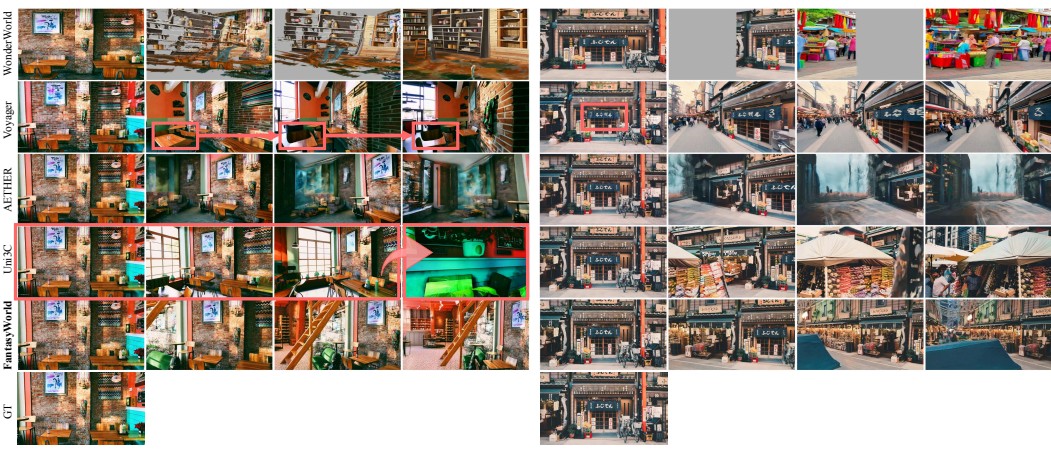

Figure 4: Qualitative comparison of world generation. WonderWorld shows missing regions, Voyager suffers from temporal incoherence and degraded first-frame fidelity, AETHER produces low-detail outputs, and Uni3C exhibits abrupt stylistic shifts. In contrast, FANTASYWORLD maintains stronger 3D consistency and coherent style across views.

## 4.3 GEOMETRIC FIDELITY

**Quantitative Comparison.** We evaluate video–geometry consistency using 3DGS (Kerbl et al., 2023) on 100 *RealEstate10K* samples, comparing three settings (Tab. 2): removing the geometry branch with VGGT initialization, our full model with VGGT initialization, and our full model with feed-forward point-cloud initialization. Results show that, under the same VGGT initialization, adding the geometry branch consistently improves PSNR/SSIM and reduces LPIPS, confirming its role in enforcing 3D consistency. Direct initialization from our predicted point clouds yields slightly lower scores than VGGT initialization, but still provides competitive results, indicating that our geometry branch produces meaningful 3D structure without relying on external supervision.

**Qualitative Analysis.** We compare reconstructed 3D scenes from Voyager, AETHER, Uni3C, and our method on indoor and outdoor cases (Fig. 5). All methods use the same pipeline: VGGT predicts point maps from generated frames, then reconstructs point clouds. Baselines show structural artifacts and duplication (e.g., blurred signage, layered walls, distorted layouts), while our method yields cleaner, more complete geometry. Indoors, walls and corners stay rectilinear and closed; outdoors, text and facades are sharper, indicating stronger 3D consistency.

Table 2: 3DGS reconstruction on *RealEstate10K*. Post-reconstruction (Post Rec) indicates the 3DGS initialization source: either from the VGGT point cloud or our own feed-forward point cloud.

| Method | Post Rec. | PSNR ↑ | SSIM ↑ | LPIPS ↓ |
|---|---|---|---|---|
| Ours w/o 3D | VGGT | 26.89 | 0.84 | 0.17 |
| Ours w/ 3D | VGGT | **28.24** | **0.86** | **0.14** |
| Ours w/ 3D | - | 26.54 | 0.85 | 0.19 |

**Ablation on the Geometry Branch.** As shown in Table 2, the same geometry-ablated variant described in Sec. 4.2 produces weaker reconstruction quality, while the full FANTASYWORLD achieves clearer geometry and thus confirms the geometry branch is crucial for robust 3D representation.

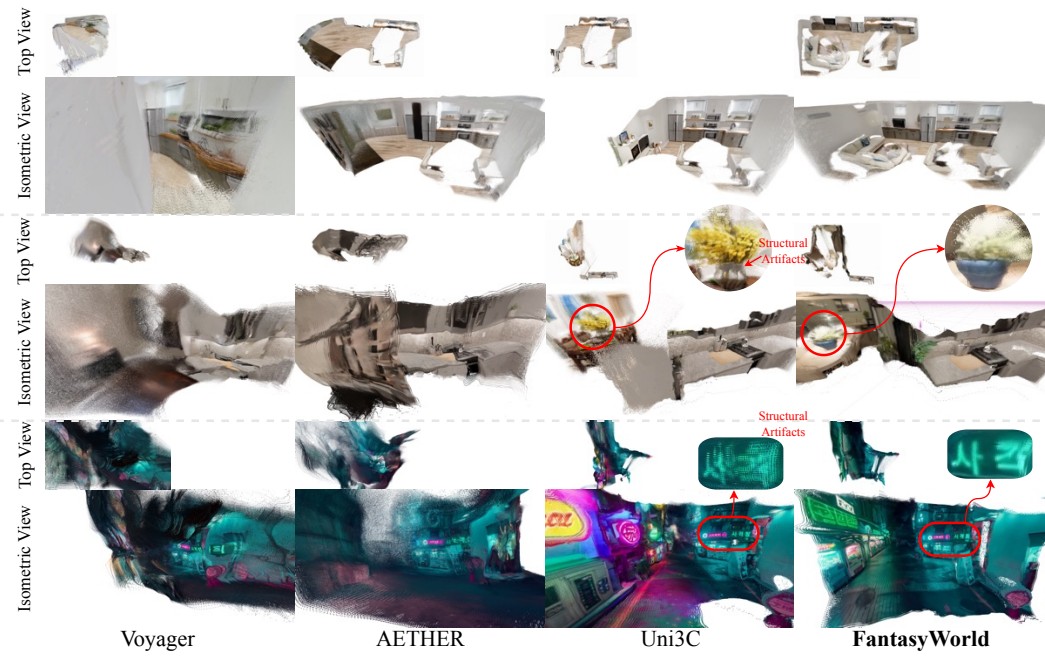

Figure 5: Qualitative Comparison of Geometry Fidelity. Top-view and isometric-view point clouds from Voyager, AETHER, Uni3C, and our method. The red-outlined regions indicate typical structural artifacts in baselines (e.g., duplicated walls, bent surfaces, blurred text), while our method preserves cleaner layouts and sharper geometry.

## 5 CONCLUSION AND LIMITATIONS

In this work, we presented FANTASYWORLD, a unified feed-forward model designed to generate 3D-consistent, explorable virtual worlds in a single pass. Our core contribution is a novel architecture that bridges the imaginative power of a pre-trained video diffusion backbone with the geometric rigor of a VGGT-style 3D geometry branch. To achieve this, we separate the prediction of 3D geometry from video appearance, introducing a dedicated geometry branch that operates on the video model's internal features while preserving its powerful generative capabilities. This is facilitated by a bidirectional cross-attention mechanism that allows video and geometry to mutually reinforce one another during generation. Our experiments show that the generated videos not only maintain strong visual realism but also achieve higher multi-view coherence and improved geometric fidelity compared to existing methods, offering a practical and efficient path toward creating structured, reusable world models for embodied AI.

Our model is currently designed for fixed-length clip generation, and extending it to continuous, long-range synthesis remains an important next step. Achieving this will require developing ef-

fective caching or streaming mechanisms for our implicit 3D representation to maintain state over time, as partially explored in recent work such as Context-as-Memory (Yu et al., 2025b). At the same time, the present study focuses on static or quasi-static scenes where geometry is approximately rigid within the generated window. Handling fully dynamic environments with strong non-rigid motion and time-varying structure remains an open limitation of our current instantiation, and will require both richer data and explicit modeling of 4D geometry, for example through time-conditioned geometry fields or deformation modules inspired by recent 4D extensions of VGGT-style backbones. Our architectural design is compatible with such extensions, but systematically validating them is left for future work. In contrast, our primary focus in this paper has been on validating the potential of large video diffusion backbones to serve as world models when paired with geometric supervision. This design choice allowed us to clearly demonstrate our central hypothesis: that an implicit 3D representation, learned directly from a video model's hidden features, is a powerful and effective tool for enforcing 3D consistency, while opening a path toward longer-horizon and fully dynamic world modeling in subsequent work.

## REPRODUCIBILITY STATEMENT

We ensure reproducibility by detailing FANTASYWORLD's architecture (PCB, IRG, bidirectional cross-attention) in Sec. 3, dataset composition and preprocessing in Sec. 4.1 and A.5, training protocols in Sec. 4.1 and A.4, and evaluation metrics, splits, results, and ablation studies in Sec. 4.1, 4.2, and 4.3. Baseline code and pretrained weights are documented in A.5, and the full codebase is available at `https://github.com/Fantasy-AMAP/fantasy-world`.

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

## A  APPENDIX

### A.1  DECLARATION ON LLM USAGE

We used a large language model (LLM) to aid in the preparation of this manuscript. Specifically, the LLM was employed to polish the writing, improve readability, and refine grammar and style. All technical content, including the model design, experimental methodology, and results, was conceived, implemented, and validated by the authors. The LLM did not contribute to the development of algorithms, the execution of experiments, or the analysis of data. Its role was limited to assisting with language clarity and presentation.

### A.2  CAMERA MOTION CONTROL

For camera motion control, we largely follow the methodology of (Wan et al., 2025), which consists of a camera pose encoder and a pose adapter. Our camera pose encoder mirrors their design, transforming camera parameters into multi-level feature embeddings via Plücker coordinates and a series of convolutional operations.

Our primary modification lies in the camera pose adapter. While the original work utilizes a full Adaptive Layer Normalization (AdaLN) to predict both scaling ($\gamma_i$) and shifting ($\beta_i$) parameters, we propose a streamlined variant that exclusively generates the shifting parameters. The features are integrated into the video latent $f_{i-1}$ at each layer $i$ through a simple additive projection:

$$f_i = f_{i-1} + \beta_i,$$

where $\beta_i$ is derived from the camera embeddings. Also the camera control module is adapt only the first 24 block among the 40 transformer blocks.

### A.3  3D DPT HEAD ARCHITECTURE

Our model employs a custom 3D DPT head, which extends the 2D version from VGGT (Wang et al., 2025a) to process video sequences via two key innovations, as shown in Fig 6.

First, we invert the spatial reassembly logic to align with the nature of diffusion backbones. Conventional DPTs, designed for standard encoders, assume that shallow layers contain high-frequency spatial details and therefore upsample them most aggressively. Diffusion models operate differently: they progressively denoise the input at each block. This means deeper blocks produce features that are not only semantically richer but also less noisy and more structurally reliable. Consequently, we invert the DPT logic. Our Geometry-Consistent Branch uses features from blocks {8,12,18,24} and upsamples features from the deepest layers the most, while downsampling those from the shallower ones. This anchors the fusion process in the highest-quality information the backbone provides.

Second, we temporally upsample each feature stream using two sequential Temporal Blocks after the reassemble block. The design of the upsample block is inspired by the WanVAE decoder (Wan et al., 2025). Each block first doubles the temporal resolution and then applies a causal 3D convolution. This process results in a total 4x temporal upsampling factor, transforming an input of $t$ frames into a smooth output sequence of $T = 4(t - 1) + 1$ frames.

### A.4  TRAINING DETAILS FOR THE GEOMETRY BRANCH

**Coordinate and Scale Normalization.** Let $E_i = [R_i \mid t_i] \in \mathbb{R}^{3 \times 4}$ denotes camera-from-world extrinsic. We align all frames to the first frame by:

$$\tilde{E}_i = E_i E_0^{-1},$$

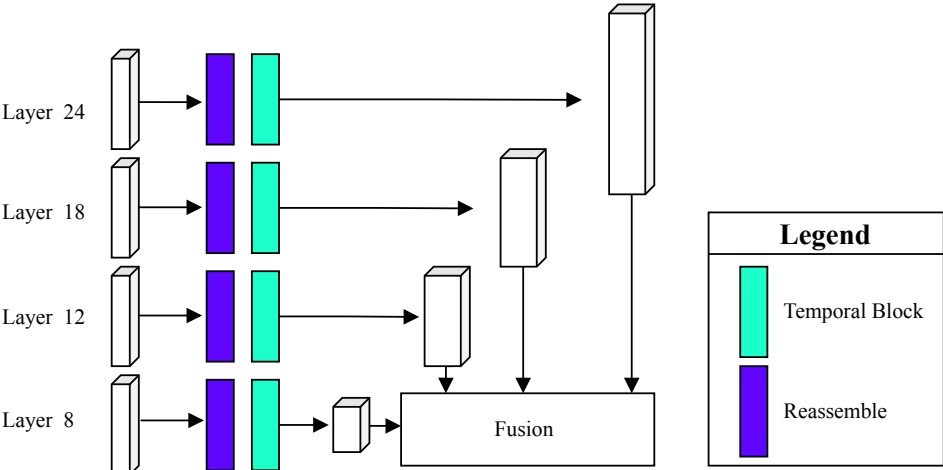

Figure 6: Our 3D DPT head with temporal blocks for temporal upsampling.

so that the first camera is the canonical reference. Let $X_{i,p}^w \in \mathbb{R}^3$ denote the world-space point at pixel $p$ of frame $i$, and let $\tilde{X}_{i,p}^w$ be its coordinates in the first frame. We further normalize the global scale using the average point radius:

$$s = \frac{1}{|\mathcal{V}|} \sum_{(i,p) \in \mathcal{V}} \left\| \tilde{X}_{i,p}^w \right\|_2,$$

where $\mathcal{V}$ is the valid-point set. The same factor is applied to extrinsic translations, depths, camera-space points, and world-space points.

In practice, some datasets provide sparse geometry or geometry concentrated on foreground objects, which makes scale estimation from GT point clouds unstable. For such cases, we recover scale from the first frame using MoGe: we reconstruct a first-frame point cloud from MoGe depth and use its average radius as the scale. This single scale is then applied to the whole sequence like extrinsic translations, depths, camera-space points, and world-space points.

**IRG Block.** IRG blocks unify the Imagination Prior Branch and the Geometry-Consistent Branch through bidirectional cross-attention. The Imagination Prior Branch produces a spatio-temporal feature grid $X_v \in \mathbb{R}^{t \times (h \times w) \times C_v}$, where $t = 21$, $h = \frac{H}{16}$, $w = \frac{W}{16}$, $H = 336$, $W = 592$, $C_v = 5120$, and the Geometry-Consistent Branch maps $X_v$ to the implicit 3D field $X_g \in \mathbb{R}^{t \times (h \times w) \times C_g}$, where $C_g = 1024$, which is then decoded into depth maps, point maps, and camera poses.

**Camera Pose Prediction.** Following VGGT (Wang et al., 2025a), we concatenate one learned camera token and four register tokens to the video token sequence. Because camera trajectories are temporally smooth in videos, we apply a lightweight 1D convolution to the camera token stream to perform temporal upsampling. For each frame $f_i$, the geometry head predicts a 9D camera parameter vector $\mathbf{g}_i \in \mathbb{R}^9$ (rotation, translation, and field-of-view).

**Depth and Point Map prediction** We use hidden features from the Geometry-Consistent Branch's layers $\{8, 12, 18, 24\}$ with shapes $\mathbb{R}^{t \times (h \times w) \times C_g}$. The features are fed a Depth 3D DPT head and a Point-Map 3D DPT head. The depth head produces $D \in \mathbb{R}^{T \times H \times W}$, and the point head produces a 3D point map $P \in \mathbb{R}^{T \times H \times W \times 3}$ together with a confidence (uncertainty) map $\Sigma^P \in \mathbb{R}^{T \times H \times W}$, where $h = \frac{H}{16}$, $w = \frac{W}{16}$, and $T = (t-1) \times 4 + 1 = 81$.

**Camera loss.** We supervise camera parameters with a robust Huber loss, where $\hat{\mathbf{g}}_i$ is the ground truth:

$$\mathcal{L}_{\text{camera}} = \sum_{i=1}^{N} \left\| \hat{\mathbf{g}}_i - \mathbf{g}_i \right\|_{\epsilon}.$$

**Depth loss.** We adapt Video Depth Anything (Chen et al., 2025b) and combine a temporal gradient matching term with a per-frame scale-sensitive spatial loss term:

$$\mathcal{L}_{\text{depth}} = \alpha\,\mathcal{L}_{\text{TGM}} + \beta\,\mathcal{L}_{\text{frame}},$$

where $\mathcal{L}_{\text{TGM}}$ enforces temporal consistency of depth gradients, and $\mathcal{L}_{\text{frame}}$ measures per-frame depth error without scale/shift normalization.

**Point-map loss.** Following VGGT ((Wang et al., 2025a)), we penalize both point positions and local gradients, weighted by the predicted uncertainty:

$$\mathcal{L}_{\text{pmap}} = \sum_{i=1}^{N} \left\| \Sigma_i^P \odot \left(\hat{P}_i - P_i\right) \right\| + \left\| \Sigma_i^P \odot \left(\nabla\hat{P}_i - \nabla P_i\right) \right\| - \gamma \log \Sigma_i^P.$$

**Total objective.** The geometry branch is trained with

$$\mathcal{L}_{\text{geo}} = \mathcal{L}_{\text{depth}} + \mathcal{L}_{\text{pmap}} + 3\,\mathcal{L}_{\text{camera}}.$$

## A.5 Evaluation Details

To ensure the reproducibility of our evaluation, we provide the implementation details of all baselines considered in our benchmark. For each method, we list the official GitHub repository, the specific commit used in our experiments, and the resolution and number of frames generated per sample.

**Repositories and Commits.**

- **Voyager:** `https://github.com/Tencent-Hunyuan/HunyuanWorld-Voyager` (commit `54a658b`)
- **Uni3C:** `https://github.com/alibaba-damo-academy/Uni3C` (commit `75ed6e2`)
- **AETHER:** `https://github.com/InternRobotics/Aether` (commit `f2221b8`)
- **WonderWorld:** `https://github.com/KovenYu/WonderWorld` (commit `5cf1146`)

**Resolution and Frame Settings.** We followed the official default inference settings of each baseline:

- **FantasyWorld (ours):** $336 \times 592$, 81 frames
- **Voyager:** $512 \times 768$, 49 frames
- **Uni3C:** $480 \times 768$, 81 frames
- **AETHER:** $480 \times 720$, 41 frames
- **WonderWorld:** $512 \times 512$, 50 frames

All baselines were run with the official inference scripts and default hyperparameters from the respective repositories.

**Efficiency and GPU Peak Memory Usage.**

In this section, we report inference footprint under a single A100-80GB GPU for various methods in Table 3, including GPU peak memory usage, inference time, number of inference frames, frame resolution, and an efficiency metric PPS (pixels per second) which defined as:

$$\text{PPS} = \frac{\text{Width} \times \text{Height} \times \text{Frames}}{\text{Inference Time (s)}}$$

As shown in the Table 3, FantasyWorld incurs a moderate overhead over Wan2.1 due to the geometry branch, remains comparable to Uni3C, and is more efficient than Voyager in terms of PPS.

## A.6 Extending controllability via post-hoc explicit reconstruction

In this section, we study how FantasyWorld can be extended with stronger camera controllability through a post-hoc explicit reconstruction step. The main paper focuses on the native controllability

Table 3: Comparison of memory usage and efficiency across methods.

| Method | GPU Memory (GB) | Efficiency (PPS) | Inference Time (s) | Inference Frames (Frames) | Resolution (W× H) |
|---|---|---|---|---|---|
| Wan2.1 | 44.91 | 44,793 | 359.70 | 81 | 592×336 |
| **FantasyWorld** | 58.24 | 33,856 | 475.90 | 81 | 592×336 |
| Voyager | 58.59 | 16,914 | 1139.13 | 49 | 768×512 |
| Uni3C | 64.51 | 37,607 | 794.00 | 81 | 768×480 |

FantasyWorld  VGGT  FantasyWorld  VGGT

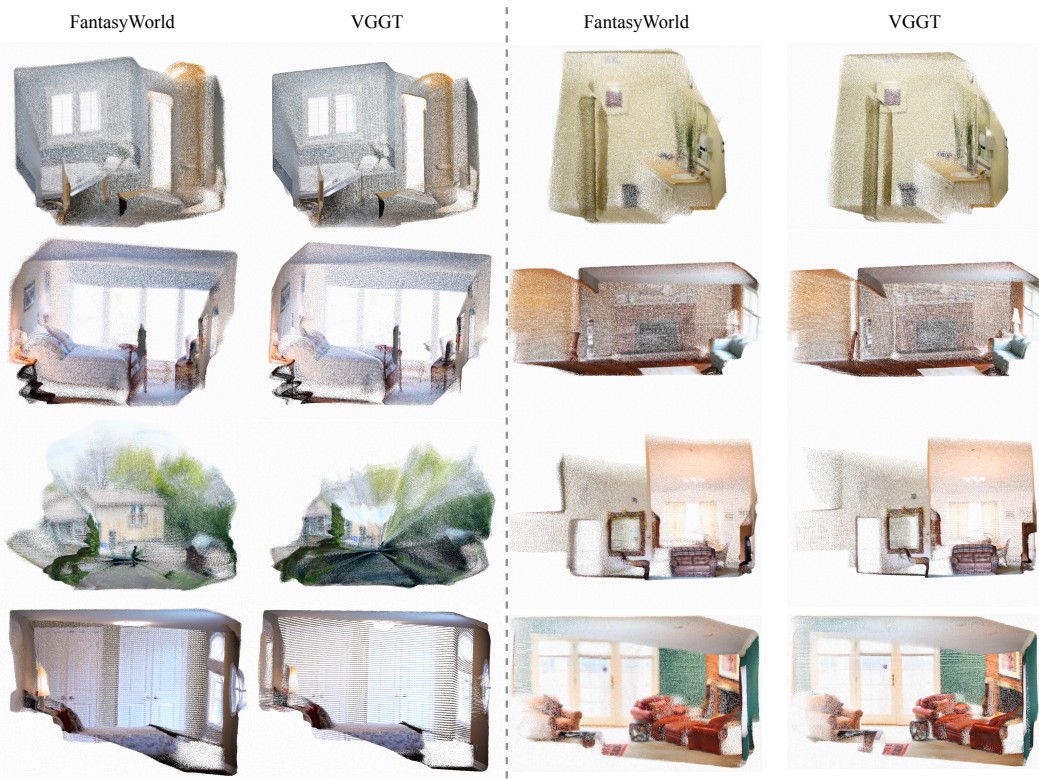

Figure 7: Comparison of feed-forward 3D predictions from FantasyWorld and VGGT. Both models produce 3D geometry in a single forward pass; FantasyWorld's video-grounded 3D branch achieves comparable reconstruction quality to VGGT.

of a single-pass video–3D model, evaluated by WorldScore. However, the proposed 3D-consistent latent representation also allows explicit control modules to be attached as plug-in components when higher controllability is required.

To illustrate this, we construct a two-stage pipeline on top of FantasyWorld. Given an input image, text prompt, and camera trajectory, we first run FantasyWorld to generate a 3D-consistent latent trajectory, and then reconstruct a Gaussian scene for each case as a post-hoc explicit reconstruction step. The reconstructed scene can subsequently be re-rendered along an arbitrary target camera trajectory. We evaluate this pipeline on 50 videos sampled from the Small Camera Motion split, which corresponds to the official WorldScore static benchmark setting used in Table 1, and report the WorldScore metrics in Table 4. These results demonstrate that, once a 3D-consistent latent representation is available, explicit control modules can be seamlessly integrated as plug-ins to obtain high camera control *without* degrading, and in fact improving, overall consistency.

We intentionally keep such reconstruction and explicit control modules outside the main model for two reasons. First, they mix the contribution of our core architecture with that of an additional

Table 4: WorldScore metrics when augmenting FantasyWorld with a post-hoc explicit reconstruction method.

| Method | 3D Consist. | Photo Consist. | Style Consist. | Camera Ctrl. |
|---|---|---|---|---|
| WonderWorld | 88.35 | 73.79 | 70.74 | **98.94** |
| FantasyWorld (post-hoc) | **88.88** | **86.59** | **90.84** | 90.71 |

reconstruction stage, which makes attribution less clear. Second, WorldScore camera and object control are designed to capture the native controllability of video models, whereas post-hoc re-rendering partially bypasses this objective. Here we focus on a simple camera-control example to illustrate the extensibility of FANTASYWORLD; exploring richer object-level control modules is a natural direction for follow-up work.

## A.7 DYNAMIC AND 4D SCENARIOS

In the main paper, FantasyWorld is presented as a unified video-3D diffusion framework for generating structurally consistent 3D worlds under camera motion. Our experiments therefore focus on predominantly static or quasi-static scenes, matching applications such as mapping, navigation, and novel-view synthesis where stable 3D structure and reliable geometry evaluation are the main goals.

Architecturally, FantasyWorld is naturally compatible with dynamic and 4D extensions. The video branch is a frozen Wan2.1 video diffusion model that encodes rich spatio-temporal information, including non-rigid motion and lighting variations, in its latent tokens. The geometry branch and IRG blocks operate on the same spatio-temporal latents via MM-BiCrossAttention, so geometry and appearance are conditioned on the full clip and are not restricted to rigid worlds by design.

To extend from static to fully dynamic scenes, one can re-parameterize the implicit field to be time-aware while keeping the IRG-based cross-branch supervision intact, for example, by using a time-conditioned geometry representation or adding a deformation representation on top of a canonical geometry to capture non-rigid motion, following VGGT-style works such as 4DGT (Xu et al., 2025). With suitable dynamic data and supervision, the unified video-3D formulation naturally generalizes to 4D settings in which scene structure, appearance, motion, and camera pose evolve over time.

## A.8 MORE EXAMPLES

For completeness, we include supplementary qualitative results covering both 3D feed-forward predictions and video generation.

Fig 7 presents additional side-by-side comparisons between the 3D geometry predicted by our geometry branch and the feed-forward baseline VGGT. Both models produce 3D structure in a single forward pass without any multi-view optimization. Across a broad range of indoor and outdoor scenes, FantasyWorld's geometry branch consistently recovers coherent spatial layouts, maintains stable global structure, and produces smooth and continuous surfaces. These examples illustrate that the geometric pathway of FantasyWorld captures strong spatial cues directly from video-conditioned features, enabling it to serve as a robust feed-forward 3D predictor within our unified video–geometry framework.

In addition to feed-forward 3D predictions, we also provide qualitative results on scenes that are far outside FantasyWorld's training distribution. These examples include stylized artwork, illustrated environments, and other non-photorealistic inputs that differ significantly from the domains seen during training. The goal of these results is to demonstrate how the model responds to inputs with unusual styles or appearances. As shown in Fig 8, the generated trajectories remain temporally stable and structurally coherent, suggesting that FantasyWorld maintains reliable video–3D representations even under substantial distribution shifts.

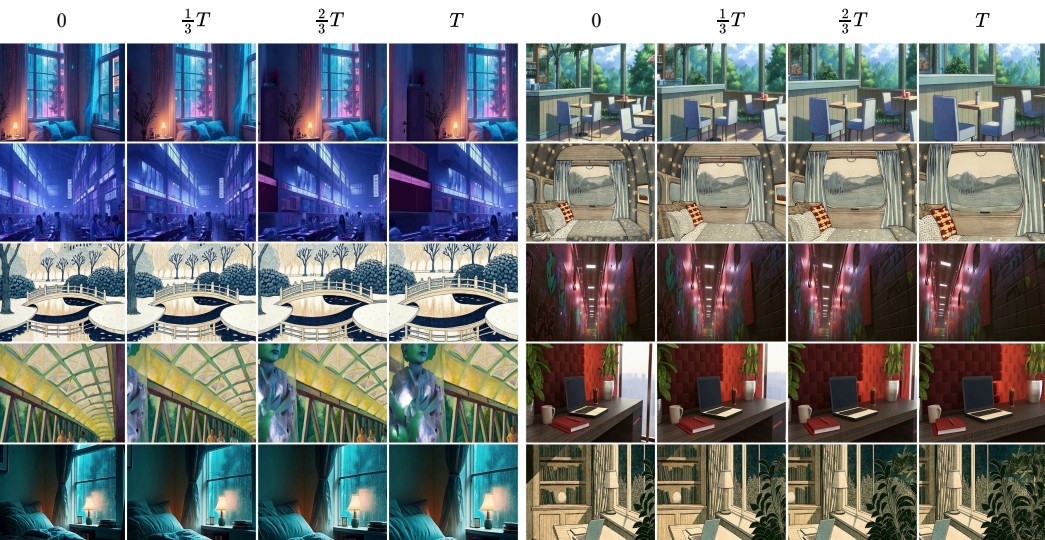

Figure 8: We present FantasyWorld's video generations on scenes far outside the training domain. Each row shows a full trajectory ($T$) sampled at four canonical timestamps. Despite substantial domain shifts, the model maintains stable appearance, coherent structure, and smooth temporal transitions throughout the sequence.

