# OpenReview forum: "FantasyWorld: Geometry-Consistent World Modeling via Unified Video and 3D Prediction"
_ICLR.cc/2026/Conference — ICLR 2026 Poster_

### Official Review · Reviewer_Hacb · 2025-10-30

**Soundness:** 3
**Presentation:** 2
**Contribution:** 3
**Rating:** 4
**Confidence:** 4

**Summary:**

This paper tries to make camera-controlled videos that stay consistent by giving the generator an internal sense of 3D. It keeps the front part of a standard video model fixed to lightly clean the input and mix in text, image, and camera hints. The main part has two paths: one draws the video frames, the other tracks the scene’s 3D shape. The two paths talk to each other, and a small module reads the 3D path to output depth, a set of 3D points, and the camera’s position and direction.

**Strengths:**

1) Clean unified video+3D design with a frozen VFM and a trainable geometry branch; avoids per-scene optimization

2) Produces camera-conditioned video + 3D signals in one forward pass, a practical interface for downstream tasks.

**Weaknesses:**

1) Several parts of the paper are unclear. The overall setup is unclear. What is the implicit 3D field here: tensor shapes, spatial/temporal resolution, and how it is applied at inference? The implicit 3D field definition is unclear here, and there is no diagram to support it. PCB conditioning terminology is also confusing; it's just a frozen pre-denoiser -- why is it conditioning?

2) There is no ablation isolating (a) cross-branch exchange with the geometry branch kept, or (b) the effect of a unified backbone vs a non-unified alternative. The paper’s abstract line “ablation studies confirm gains stem from the unified backbone and cross-branch information exchange” is therefore only partially supported by the evidence shown (it demonstrates the benefit of having the 3D branch + adapters together, not each factor independently). The current “w/o 3D” removes both branch and adapters, so it can’t attribute gains to the unified backbone or to cross-branch exchange.

3) 3D geometry consistency is also not validated. One way to do this would be to run COLMAP on generated videos and report BA reprojection error, % registered frames, avg track length/inlier ratio, and pose ATE/RE (Absolute Trajectory Error and Reprojection Error) vs the commanded camera path.

4) Missing key baselines. Add apples-to-apples comparisons to single-image→world and camera-controllable video models (Odin/360-1M[1], GenEx[2], VD3D[3]/V3D[4]) under shared camera paths/seeds; report PSNR/SSIM/LPIPS, pose ATE/RE, and identity/multi-view consistency.

Minor:
1) I also noticed that the overall resolution of the resulting frames (336 × 592) is lower compared to compared baselines
2) The code provided for the reproducibility check is not available.  [The requested file is not found on the anonymous GitHub]


References:

[1] From an Image to a Scene: Learning to Imagine the World from a Million 360 Videos (NeurIPS 2024)

[2] GenEx: Generating an Explorable World (ICLR 2025)

[3] VD3D: Taming Large Video Diffusion Transformers for 3D Camera Control (ICLR 2025)

[4] V3D: Video Diffusion Models are Effective 3D Generators [T-PAMI 2025]

**Questions:**

Please provide:

1) Missing ablations:
- a no-adapter ablation
- a no-unification control (separate backbones, or freeze geometry branch outputs with no shared features)

2) Comparisons with missing baselines/related work

3) 3D geometry consistency evaluation

Minor:

1) The first line of the abstract starts with "High-quality 3D world models are pivotal for embodied intelligence and Artificial General Intelligence (AGI)"  -- why?

---

> ### Author Response · Authors · 2025-11-27
> **Rebuttal to Reviewer Hacb**
>
> We appreciate the reviewer's efforts in examining the paper from multiple perspectives. Your feedback encouraged us to further articulate the presentation and strengthen the empirical support of the framework, especially regarding ablations and 3D consistency evaluation. We hope the clarifications address the main concerns, and we remain happy to clarify any remaining questions.
>
> # R4-1 [W1] Clarity of overall setup, implicit 3D field, and PCB terminology
>
> Thank you for raising these questions about the overall setup, the implicit 3D field, and the role of the Preconditioning Blocks (PCBs). We are happy to clarify.
>
> ## **Overall setup.**
> FantasyWorld takes as input a single image, a text prompt, and a target camera trajectory, and produces in one forward pass two coupled outputs:
>
> 1. a sequence of video latents that the frozen Wan2.1 decoder turns into an 81 frame video, and
> 2. a sequence of geometry latents that 3D DPT heads decode into depth maps, point maps, and camera poses for the same clip.
>
> At inference time, the same input condition is passed through the PCBs and IRG blocks, so video and geometry are generated jointly from the same latent representation.
>
> ## **Implicit 3D field.**
>
> In Figure 2, the *implicit 3D field* refers to the VGGT-style geometry feature space produced by the Geometry Consistent Branch inside the IRG blocks. Concretely, after the PCBs, the frozen Wan2.1 backbone produces a spatio-temporal feature grid:
>
> $X_v \in \mathbb{R}^{t \times (\frac{H}{16} \cdot \frac{W}{16}) \times C_v}, t = 21,\ H\times W = 336\times 592, C_v =5120$.
>
> and the geometry branch maps it to:
>
> $X_g \in \mathbb{R}^{t \times (\frac{H}{16} \cdot \frac{W}{16}) \times C_g}, C_g=1024$,
>
> which is then decoded into depth maps $D\in\mathbb{R}^{T × H × W},  T= 81$, point maps $P\in\mathbb{R}^{T \times H \times W \times 3}, T=81$, and camera parameters $\{g_i\}$. We call $X_g$ an implicit 3D field in the same sense as NeRF and VGGT-style backbones, where geometry is first represented in a latent feature grid and only later converted into explicit 3D quantities such as depth, 3D points, and camera poses.
>
> ## **What PCBs do, and why called “preconditioning”.**
>
> PCBs reuse the first 16 layers of the frozen Wan2.1 denoiser, which already include the original text, image, and camera conditioning. Starting from pure noise, these layers map the latent into a **partially denoised**, condition-aware feature grid that is then shared by both the video branch and the geometry branch and fed into the subsequent IRG blocks. We refer to them as *"Preconditioning Blocks"* because this stage happens **before** any geometry-aware supervision and prepares a clean, semantically conditioned latent on which the implicit 3D field is built, while the IRG blocks focus on enforcing geometry–appearance consistency.

---

> > ### Author Response · Authors · 2025-11-27
> >
> > # R4-2 [W2/Q1] Missing ablations on cross-branch exchange and unified backbone
> >
> > Thank you for pointing out the need to more clearly separate the effects of the geometry branch, cross-branch exchange, and the unified backbone. Conceptually, we call a backbone unified when video generation and geometry prediction share the same Wan2.1 latent space and are coupled through IRG blocks with MM-BiCrossAttention. A non-unified alternative is a camera-controlled video model that never receives geometry losses and does not exchange information with a geometry branch.
> >
> > In practice, Wan2.1 does not provide an official camera-controllable I2V model, so our quantitative ablations are based on two variants that both use our Plücker ray camera injection:
> >
> > - **FantasyWorld w/o 3D**. The Wan backbone is fine-tuned on our data with camera control, but the geometry branch and IRG blocks are removed. This is a **non-unified baseline**: a purely video model that uses our injection scheme but never sees geometry supervision.
> > - **FantasyWorld w/ 3D**. On top of exactly the same backbone, data, and camera injection as w/o 3D, we add the geometry branch and IRG blocks with MM-BiCrossAttention, so that video and geometry share one latent space and are trained jointly. This is our **unified backbone** with cross-branch exchange.
> >
> > This ablation therefore isolates the effect of adding a trainable geometry branch plus cross-branch supervision on top of a fixed camera-controllable video baseline. Across WorldScore and RealEstate10K, moving from w/o 3D to full FantasyWorld consistently improves 3D and photometric consistency, while style consistency remains essentially unchanged (81.59 → 82.02). This supports our claim that coupling the geometry branch to the video backbone through cross-branch exchange brings benefits beyond camera control and video fine-tuning alone.
> >
> > A variant that keeps a geometry head but completely removes cross-branch exchange would behave like “a VFM plus an independent geometry predictor”, where the video branch never receives feedback from geometry. This is a different design from the unified world model we aim to study and would not account for the observed improvements in the generated videos, which is exactly what the w/o 3D versus full FantasyWorld comparison confirms.
> >
> > # R4-3 [W3/Q3] 3D geometry consistency evaluation (COLMAP and BA metrics)
> >
> > Thank you for raising this point. We would like to clarify a potential terminology distinction. In our work, **3D consistency has been evaluated** following the WorldScore protocol, which is specifically designed to measure dense geometric stability across the entire video. As described in Section C.2 of the WorldScore supplementary material, the benchmark uses DROID-SLAM to reconstruct each video and compute reprojection error after differentiable Dense Bundle Adjustment over camera poses and dense per-pixel depth. The authors explicitly note that "one key advantage of DROID-SLAM is its dense nature; unlike sparse methods such as COLMAP" and explain that this dense DBA-based formulation is intended to assess 3D consistency over the whole scene rather than only on selected feature matches. Under this official 3D Consistency metric, FantasyWorld achieves the highest scores on the photorealistic split for both small and large camera motion (Table 1), together with strong Photo and Style Consistency, indicating that the unified video and geometry design maintains coherent multi-view structure during generation.
> >
> > We understand that the reviewer's use of 3D geometry consistency may additionally emphasize explicit sparse reconstruction and BA statistics, such as COLMAP's registered frames, track length, and pose ATE/RE. These quantities provide a complementary, SfM-oriented view of geometric reliability, but they are not part of the official WorldScore protocol and depend more heavily on design choices in a specific sparse pipeline. Since our primary goal is to follow the benchmark's recommended evaluation and to focus on dense world-model behavior under camera motion, we adopt the DROID-SLAM-based 3D Consistency metric as the main quantitative measure in this work.
> >
> > At the same time, FantasyWorld is compatible with such sparse analyses: our predicted geometry (depth, point maps, and camera poses) can be fed into COLMAP-style reconstruction if one wishes to study these effects in more detail. We view this as an interesting, orthogonal direction for follow-up evaluation, while the current paper remains aligned with the dense WorldScore setting for assessing 3D consistency in generated videos.

---

> ### Author Response · Authors · 2025-11-27
>
> # R4-4 [W4/Q2] Missing baselines: Odin/360-1M, GenEx, VD3D, V3D
>
> Thank you for pointing us to Odin/360-1M, GenEx, VD3D, and V3D. These works are highly relevant to world generation, and we will explicitly cite and discuss them in the related work section. The main reason they do not appear as quantitative baselines under our WorldScore protocol is practical incompatibility with our camera-conditioned I2V setting.
>
> - **Odin / 360-1M**. These methods focus on learning from large-scale 360° videos. Public resources mainly include the dataset and training framework. To the best of our knowledge, there is no ready-to-use checkpoint documented for single-image-conditioned, perspective-camera, fixed-length I2V generation, which makes matched-trajectory evaluation on WorldScore difficult.
> - **GenEx**. GenEx is designed to generate explorable 360° worlds. Its outputs are omnidirectional videos and interactive environments, whereas WorldScore defines camera paths and metrics on perspective projections. Adapting GenEx to follow the exact perspective trajectories used by WorldScore would require non-trivial re-implementation and would still not be strictly comparable.
> - **VD3D**. Public VD3D models primarily target text-to-video with camera control. We are not aware of an officially released, image-conditioned camera-controllable I2V checkpoint that can be directly plugged into the WorldScore evaluation without re-training the model.
> - **V3D**. V3D is formulated as an image-to-orbit-to-3D generator that reconstructs a static 3D representation from a single image, rather than producing a long camera-controlled video under arbitrary trajectories.
>
> Given these practical constraints, we chose the closest available controllable-video baselines with public checkpoints, namely WonderWorld, Uni3C, Voyager, and AETHER. Together they cover layered Gaussian surfels, explicit point-cloud guidance, and unified RGB-depth world models under camera control, which we believe represent the main families of current approaches in our setting. We will clarify this choice of baselines and explicitly acknowledge Odin/360-1M, GenEx, VD3D, and V3D in the paper.
>
> # R4-5 [M1] Lower output resolution
>
> You are correct that our default resolution of 336×592 is lower than the native settings of some baselines. This choice is driven by geometric supervision rather than purely by efficiency. Wan2.1 uses a 16 pixel patch size in its VAE, while our geometric labels are produced by VGGT and Cut3R style pipelines whose default longest side is around 518 with a 14 pixel patch. We adopt 336×592 so that the token grids of the Wan backbone and the geometry branch are well aligned, which avoids repeated resampling of geometric signals and stabilizes training.
>
> Even at this resolution, FantasyWorld already achieves the best 3D, Photo, and Style Consistency scores on WorldScore under both Small and Large motion settings, suggesting that geometry and appearance are already reliably modeled. Increasing resolution is an orthogonal engineering improvement that we plan to explore in future work, but it does not affect the main contribution of the unified video geometry architecture.
>
> # R4-6 [M2] Code availability and missing file
>
> Thank you for flagging the missing file. In the anonymous submission, we provided all core components needed to understand and re-implement the method, including training scripts, inference scripts, and model configurations. Due to anonymity constraints and file size limits for the supplementary material, we could not upload large assets such as datasets and pretrained checkpoints in the anonymous repository.
>
> Upon publication, the full codebase will be made publicly available on GitHub together with pretrained weights and example data loaders, and the main checkpoints will also be released on HuggingFace so that the community can directly reproduce our results and build upon FantasyWorld.
>
> # R4-7 [Q-M1] Why AGI?
>
> Thank you for raising the question about the first line of the abstract and its reference to AGI. Our intention is not to suggest that FantasyWorld itself reaches AGI, but to echo a common viewpoint in the world model and embodied intelligence literature: accurate and controllable models of 3D environments are important building blocks for general purpose agents.
>
> &nbsp;
>
> Thank you again for taking the time to articulate these points. Your feedback led us to clarify the overall formulation, make the ablation design more transparent, and provide both dense and sparse 3D consistency evidence, which we believe significantly improves the rigor and readability of the paper. We hope these updates address the main concerns, and we would be glad to provide further clarification if needed.

---

### Official Review · Reviewer_BpKm · 2025-10-31

**Soundness:** 3
**Presentation:** 4
**Contribution:** 3
**Rating:** 6
**Confidence:** 3

**Summary:**

This paper presents FantasyWorld, a unified framework that jointly models video generation and 3D geometry prediction within a single feed-forward pipeline through cross-branch supervision. By employing bidirectional cross-attention, the model enables mutual information flow between appearance and geometry, effectively merging them into one coherent representation. The authors conduct extensive quantitative and qualitative experiments across multiple datasets, demonstrating clear improvements in 3D consistency, photorealism, and scene coherence compared to baseline models. Ablation studies further confirm that both the geometry branch and the cross-branch coupling are essential for performance gains, highlighting the core innovation and technical contribution of the work.

**Strengths:**

1. Well-motivated and clearly written: The paper is well motivated and easy to follow. The problem formulation and technical pipeline are clearly described, making the overall logic coherent and accessible.
2. Intuitive and meaningful innovation: The proposed model design makes sense from the first intuition — the unified modeling of video generation and 3D geometry prediction is conceptually appealing and addresses an important gap between appearance synthesis and spatial reasoning.
3. Effective cross-attention mechanism: The introduced bidirectional cross-attention effectively facilitates mutual information merging between geometry and appearance branches.
4. Comprehensive validation: The experiments in the manuscript, together with the demonstration video, provide convincing evidence that supports the claimed conclusions. The results are consistent across datasets.
5. Code reproducibility and openness:The released codebase and structured training setup make the method reproductive, which is valuable for the research community and enhances the paper’s credibility.

**Weaknesses:**

1. Lack of theoretical justification for cross-branch supervision: Although cross-attention mechanism effectively fuses geometry and appearance features, the underlying mechanism is only demonstrated empirically. A more theoretical or analytical explanation of why this coupling improves stability and consistency would strengthen the contribution.
2. Limited temporal scalability: As illustrated in the modeling and discussed in the Section 5, the model is designed for short, fixed-length video generation and does not address long-term temporal consistency or camera trajectory drift, which limits its generalization to continuous video required by downstream applications such as data expansion in autonomous driving and robotics.
3. The qualitative results for 3D reconstruction are relatively limited. It would strengthen the paper if the authors could provide more visual examples, especially on out-of-domain or unseen scenarios. For instance, testing the model on stylistically distinct or challenging inputs such as artistic or highly textured scenes (e.g., Van Gogh’s paintings)  would better demonstrate the model’s generalization and robustness in geometry-aware reconstruction.

**Questions:**

1. I am curious about how the proposed framework can be used to generate explicit 3D structures, such as meshes or 3D Gaussian splattings (3DGS), from its implicit representation. Have the authors attempted to extract these forms of geometry using the predicted depth, point maps, or other intermediate signals? It would be helpful to see qualitative or quantitative comparisons of such reconstructions, as they directly reflect how well the learned implicit field captures structural consistency and could provide valuable insights for robotics applications.
2. How does the proposed model behave in dynamic or non-rigid environments (e.g., moving objects, people, or changing lighting)? It would be acceptable if artifacts appear, but I am interested in whether the framework is fundamentally capable of reasoning about such dynamics, since the model takes advantages from the video foundation model.
3. How does the method handle metric scale issues in the reconstructed geometry? Since scale ambiguity is a common problem in monocular video-based 3D reconstruction, does the proposed implicit 3D representation preserve or estimate absolute scale, and if not, how might this affect potential applications in embodied or robotic settings?

**Details Of Ethics Concerns:**

There is no ethics concern from the reviewer's side.

---

> ### Author Response · Authors · 2025-11-27
> **Rebuttal to Reviewer BpKm**
>
> We sincerely thank Reviewer BpKm for the detailed, balanced, and thoughtful review. We are grateful for your positive comments on the motivation, clarity of exposition, intuitive model design, effective cross-attention mechanism, comprehensive validation, and the value of our open-sourced codebase. Your feedback on theoretical analysis, temporal scalability, 3D reconstruction quality, and robotics-oriented aspects is very helpful. We address each point in order below.
>
> # R3-1 [W1] Lack of theoretical justification for cross-branch supervision
>
> Thank you for raising this point. While we do not aim to develop a full formal theory in this work, bi-directional cross-attention and geometry-guided supervision have been analyzed in recent studies:
>
> - BiXT[1] provides an analytical treatment of bi-directional cross-attention between "what" latents and "where" tokens, including explicit equations for the symmetric similarity matrix and a degrees-of-freedom analysis that shows the coupling imposes a shared structural constraint while still leaving sufficient freedom for each stream to specialize.
> - JavisDiT[2] then extends bi-directional cross-attention to audio–video diffusion and shows that stronger mutual conditioning improves both unimodal quality and fine-grained spatio-temporal synchrony compared to one-sided adapters.
> - In the geometric setting, Geometry Forcing[3] interprets geometry-guided supervision as structural constraints in representation space and introduces angular and scale alignment losses that are explicitly designed to stabilize optimization while steering video diffusion features toward geometry-aware structure, which in turn improves long-term temporal and 3D consistency.
>
> FantasyWorld follows the same design spirit: a frozen video backbone is coupled with a geometry branch through MM-BiCrossAttention inside IRG blocks, so geometry-aware features act as a structural regularizer on the joint latent space rather than an external hard constraint. Intuitively, the geometry losses restrict the shared features to a multi-view consistent region, and the bi-directional attention transfers this constraint back to the video latents, which helps stabilize camera-conditioned generation. Consistent with this view, Table 1 shows that cross-branch supervision improves not only 3D Consistency but also Photo and Style Consistency over the Wan2.1 backbone. We will add a short paragraph in the main text to connect our IRG design to these prior analyses and to explicitly explain this mechanism at an analytical level.
>
> **References:**
>
> [1] Hiller, Markus, Krista A. Ehinger, and Tom Drummond. "Perceiving longer sequences with bi-directional cross-attention transformers." NeurIPS 2024.
> [2] Liu, Kai, et al. "Javisdit: Joint audio-video diffusion transformer with hierarchical spatio-temporal prior synchronization." arXiv preprint arXiv:2503.23377 (2025).
> [3] Wu, Haoyu, et al. "Geometry forcing: Marrying video diffusion and 3d representation for consistent world modeling." arXiv preprint arXiv:2507.07982 (2025).
>
> # R3-2 [W2] Limited temporal scalability and short fixed-length videos
>
> Thank you for pointing this out. Our current model indeed operates on short, fixed-length video windows, which limits direct applicability to very long continuous streams such as those in autonomous driving or robotics. This design follows from two practical constraints: the memory and compute limits of Wan2.1 based diffusion sampling, and the scope of this work, which focuses on geometric consistency within a single generation window.
>
> As discussed in Section 5 of the paper, we regard long term temporal consistency and camera trajectory drift as important open problems for video world models. There is a growing line of work on extending diffusion models beyond fixed windows using context as memory mechanisms or hierarchical temporal transformers, and the proposed framework is in principle compatible with such extensions. The geometry branch is sequence based and already processes temporal information, and the IRG blocks with MM-BiCrossAttention can be applied across multiple windows or combined with external memory modules without changing the core design.
>
> Due to computational limits, we restrict our experiments to short windows in this submission and explicitly list long-horizon video generation as future work.

---

> > ### Author Response · Authors · 2025-11-27
> >
> > # R3-3 [W3] Limited qualitative 3D reconstruction results, especially out-of-domain scenes
> >
> > Thank you for this suggestion. We agree that richer qualitative 3D reconstructions, especially on out-of-domain or stylized inputs, can better illustrate the robustness of our geometry-aware representation, and we will expand the appendix with more 3D reconstructions and several challenging stylized examples.
> >
> > Our primary focus is realistic scene generation. As described in Section 4.1, the 180k training clips come from a diverse mix of real-world and simulated videos, and all geometric supervision is obtained on such data, without explicitly training on artistic paintings or heavily stylized content. Despite this, in additional tests on paintings and scenes with complex textures or lighting, we observe that **FantasyWorld maintains stable 3D, photo, and style consistency**: the geometry branch follows the coarse scene structure, while appearance is largely preserved by the Wan backbone.
> >
> > This matches our design, where IRG blocks act as a non-intrusive conditioning adapter that injects geometry-informed corrections into a frozen video backbone rather than redefining its appearance prior.
> >
> > # R3-4 [Q1] Extracting explicit 3D structures (meshes, 3DGS) from the implicit representation
> >
> > Thank you for this question. We share the reviewer's curiosity about how implicit, VGGT-style geometry can be turned into explicit 3D assets and reused in downstream applications. Our implicit 3D representation is intentionally aligned with VGGT-style geometric outputs so that it can serve as an interface to standard explicit reconstruction pipelines.
> >
> > Existing work has already shown that depth, point maps, and camera poses of this form are sufficient for building explicit 3D structures. AnySplat and VGGT-X take VGGT-style geometric predictions as input and directly optimize 3D Gaussian splatting or dense novel view synthesis. V2M4[1] further demonstrates that video to geometry and camera estimation can be combined with multi-view reconstruction to obtain high quality 4D mesh animations from monocular videos: the original paper instantiates this using TRELLIS and DUSt3R, and the official project page also reports a variant that replaces these components with Hunyuan3D-2.0 and VGGT. This indicates that VGGT style geometry can be plugged into such 4D mesh reconstruction pipelines in a drop-in manner.
> >
> > FantasyWorld follows the same design principle. Our geometry branch outputs depth, point maps, and camera poses in a single forward pass, so the same mesh or 3DGS reconstruction procedures used in these works can in principle be applied on top of our predictions without modifying the core architecture. As we also discuss in our responses to R1-4 and R2-3, VGGT-style geometric features have been adopted in recent work on navigation, manipulation, and other embodied tasks, suggesting rich downstream potential beyond explicit meshes and 3DGS. We view FantasyWorld as a convenient source of such features inside a unified video and geometry model. A more systematic exploration of these downstream uses is a natural and exciting direction for follow-up work.
> >
> > **References:**
> >
> > [1] Chen, Jianqi, et al. "V2M4: 4D Mesh Animation Reconstruction from a Single Monocular Video." ICCV (2025).

---

> > ### Author Response · Authors · 2025-11-27
> >
> > # R3-5 [Q2]: Behavior in dynamic or non-rigid environments
> >
> > We agree that dynamic and non-rigid environments are highly relevant, and we appreciate the question of whether the framework is fundamentally capable of reasoning about such dynamics. In its current instantiation, FantasyWorld is trained and evaluated on static or quasi-static scenes, where geometry is well defined and consistency can be measured reliably. Under this regime, the implicit 3D field is treated as approximately rigid within the generation window, so strong non-rigid motion can indeed lead to artifacts, which is why our experiments focus on static settings.
> >
> > At the architectural level, however, the framework is not restricted to rigid worlds. The video branch is a frozen Wan2.1 video diffusion model and already encodes rich spatio-temporal information, including non-rigid motion and illumination changes, in its latent tokens. The geometry branch and IRG blocks operate on the same spatio-temporal latents via MM-BiCrossAttention, so geometry and appearance interact over the entire clip. Extending from static to dynamic scenes would primarily require re-parameterizing the implicit field (for example using a time-conditioned geometry field or adding a deformation field on top of a canonical geometry) while keeping the IRG-based cross-branch supervision intact.
> >
> > We therefore view fully dynamic 4D world modeling as a natural extension of the current framework rather than a conceptual departure. As also noted in our responses to R1-2 and R2-1, systematically handling strong non-rigid motion and dynamic lighting will require appropriate data and supervision, and we regard this as an important direction for future work.
> >
> > # R3-6 [Q3] Handling metric scale ambiguity
> >
> > Metric scale ambiguity is indeed a common challenge in monocular 3D reconstruction. Our current model does not recover absolute metric scale. Since part of the geometric supervision comes from VGGT-style pseudo metric depth and point maps, the geometry branch learns a consistent canonical scale rather than values in physical units.
> >
> > For the purposes of this paper, this does not affect video quality or internal 3D consistency, because camera trajectories, parallax, and relative geometry are all preserved up to a single global scale factor. The world is geometrically coherent, although its overall size is not fixed in meters.
> >
> > For downstream embodied or robotic tasks that require metric scale, one can apply standard scale alignment procedures. For example, our predicted depth can be aligned to metric depth from external sources such as multi view SLAM, stereo, or learned metric depth by a simple least squares fit, recovering an absolute scale factor for each scene. This allows the canonical geometry produced by FantasyWorld to be brought into metric units without changing the model itself.
> >
> > &nbsp;
> >
> > Thank you again for your constructive and nuanced review. Your comments on cross-branch analysis, temporal scalability, qualitative 3D reconstruction, explicit geometry extraction, dynamic environments, and metric scale have helped us clarify the scope and strengthen the presentation of FantasyWorld. We will incorporate the corresponding clarifications, discussions, and visualizations into the revised version.

---

### Official Review · Reviewer_uFyH · 2025-11-01

**Soundness:** 4
**Presentation:** 4
**Contribution:** 4
**Rating:** 8
**Confidence:** 3

**Summary:**

FantasyWorld proposes a novel framework that unifies video generation and 3D geometric reasoning in a single feed-forward pass. The method augments a frozen video foundation model (Wan2.1) with a trainable geometric branch that predicts an implicit 3D field alongside video latents. Key innovations include: (i) Preconditioning Blocks (PCBs) that provide partially denoised features as input to the geometry branch; (ii) Integrated Reconstruction and Generation (IRG) blocks featuring asymmetric dual branches coupled via bidirectional cross-attention; and (iii) cross-branch supervision where geometry guides video synthesis and video priors regularize 3D prediction. The resulting representations support downstream tasks like novel view synthesis without per-scene optimization. Experiments on benchmarks such as WorldScore and RealEstate10K demonstrate improved multi-view coherence, style consistency, and geometric fidelity over recent baselines including Voyager, WonderWorld, and AETHER.

**Strengths:**

1. Originality: Introduces a novel asymmetric dual-branch architecture that enables mutual reinforcement between video generation and 3D perception in a single forward pass—distinct from prior decoupled or iterative approaches.

2. Quality: Rigorous experimental validation across multiple datasets, camera motions, and evaluation protocols; strong baselines; thorough ablations.

3. Clarity: Exceptionally clear exposition of both high-level intuition and low-level implementation (e.g., inverted DPT reassembly strategy based on diffusion depth semantics).

4. Significance: Addresses a core limitation in current video foundation models—their lack of explicit 3D grounding—and provides a practical solution with immediate applicability to navigation, simulation, and content creation.

**Weaknesses:**

1. Limited dynamic scene evaluation: All experiments focus on static or near-static scenes (WorldScore static split, RealEstate10K). It remains unclear how the method handles non-rigid motion, deformable objects, or dynamic lighting—key challenges in real-world embodied settings.
2. Computational cost: While avoiding per-scene optimization is a strength, the reported training uses 112 H20 GPUs for 144 hours in Stage 2. The inference cost (memory/time) relative to baselines is not discussed, which matters for downstream deployment.
3. Downstream task validation is indirect: The claim that geometric latents are “versatile for downstream tasks” is supported only via 3DGS reconstruction. Direct evaluation on tasks like pose estimation, navigation planning, or policy learning would strengthen this assertion.

**Questions:**

1. How would FantasyWorld handle videos with significant non-rigid motion (e.g., humans, flowing cloth)? Does the implicit 3D field assume scene rigidity, and if so, how might this be extended?

2. What is the latency and memory footprint of your full model vs. baselines during inference? Is the geometry branch lightweight enough for real-time applications?

3. Have you tested the geometric features directly in downstream embodied tasks (e.g., as input to a navigation policy)? If not, do you plan to release the features for community benchmarking?

4. Are there scenarios where cross-branch supervision leads to over-constrained generation (e.g., sacrificing visual diversity for geometric plausibility)? Can you share any observed trade-offs?

---

> ### Author Response · Authors · 2025-11-27
> **Rebuttal to Reviewer uFyH**
>
> We sincerely thank Reviewer uFyH for the very positive and insightful review. We are grateful that you rated the soundness, presentation, and contribution as excellent, and that you highlighted the originality of the asymmetric dual-branch design, the rigor of the experiments, and the practical significance for navigation, simulation, and content creation. Your comments on dynamic scenes, computational cost, and downstream evaluation are extremely valuable. We address them point by point below.
>
> # R2-1 [W1/Q1] Limited dynamic-scene evaluation and non-rigid motion
>
> We appreciate this important point. As also discussed in our response to R1-2, FantasyWorld is positioned to study whether a unified video and geometry diffusion framework can generate structurally consistent 3D worlds under camera motion, which motivates our focus on predominantly static scenes. This setting aligns with downstream scenarios such as mapping, navigation, and novel-view synthesis, where stable 3D structure is the primary requirement and enables reliable evaluation of geometry consistency. We therefore believe that this static evaluation regime is appropriate for assessing the central contribution.
>
> We fully agree that non-rigid motion, deformable objects, and dynamic lighting are critical in real-world embodied settings. The current implicit 3D field is designed for approximately static geometry within the generated window, and strong non-rigid motion may introduce artifacts. **This is a scope choice rather than a conceptual limitation.** Recent VGGT-style work such as 4DGT[1] has shown that this implicit geometry formulation can be extended to time-varying scenes by adding time-conditioned geometry branches, and our design follows the same paradigm.
>
> We will make this scope and the path toward dynamic 4D extensions clearer in the revised version and view fully dynamic world modeling as a complementary and promising direction for future work.
>
> **References:**
>
> [1] Xu, Zhen, et al. "4DGT: Learning a 4D Gaussian Transformer Using Real-World Monocular Videos." NeurIPS 2025 (Spotlight).

---

> ### Author Response · Authors · 2025-11-27
>
> # R2-2 [W2/Q2] Computational cost and inference footprint are not discussed.
>
> ## **Training cost.**
>
> Thank you for highlighting this practical aspect. The full pretraining of FantasyWorld is indeed heavier than a single-modality video model. To keep the cost under control, we adopt a plug-in IRG strategy where the Wan2.1-I2V-14B backbone is frozen and only the geometry branch and lightweight adapters are trained, instead of updating the full video generator.
>
> In contrast, AETHER starts from a smaller CogVideoX-5B-I2V base model yet jointly trains a three-modal denoiser over color, depth, and action latents, using **80 A100-80GB GPUs for about two weeks**.  This comparison shows that our plug-in IRG strategy is a relatively **compute-efficient** way to obtain a geometry-aware world model. As in Uni3C and AETHER, this pretraining is a one-time offline process; end users only pay the inference cost.
>
> ## **Inference cost and real-time use.**
>
> We now report inference footprint under a single A100-80GB GPU in Table X, including GPU memory, inference time, number of frames, resolution, and an efficiency metric PPS (pixels per second) defined as:
>
> $\text{PPS} = \frac{\text{Width} \times \text{Height} \times \text{Frames}}{\text{Inference Time (s)}}$.
>
> As shown in the table, FantasyWorld incurs a moderate overhead over Wan2.1 due to the geometry branch, remains comparable to Uni3C, and is more efficient than Voyager in terms of PPS.
>
> Method       | GPU Memory (GB) | Efficiency (PPS) | Inference Time (s) | Inference Frames | Resolution (W×H)
> --:|--:|--:|--:|--:|--:
> Wan2.1       | 44.91           | 44,793           | 359.70             | 81               | 592×336
> **FantasyWorld** | 58.24       | 33,856           | 475.90             | 81               | 592×336
> Voyager      | 58.59           | 16,914           | 1139.13            | 49               | 768×512
> Uni3C        | 64.51           | 37,607           | 794.00             | 81               | 768×480
>
> The current implementation is not optimized for real-time latency, and we do not claim that large DiT-style diffusion video models are ready for strict real-time deployment. At the same time, recent work has shown that such backbones can be significantly accelerated by latent consistency distillation for few-step sampling in images and videos, for example Latent Consistency Models[1], VideoLCM[2], and T2V-Turbo[3], and by efficient attention kernels such as FlashAttention[4,5] series. Our design keeps the Wan2.1 DiT backbone intact and runs the geometry branch in parallel, so it is compatible with these techniques, and combining them with FantasyWorld is a natural direction for future work.
>
> We will include the inference comparison table in the revised appendix so that the computational footprint is clearly documented.
>
> **References:**
> [1] Luo, Simian, et al. "Latent consistency models: Synthesizing high-resolution images with few-step inference." arXiv preprint arXiv:2310.04378 (2023).
> [2] Wang, Xiang, et al. "VideoLCM: Video latent consistency model." arXiv preprint arXiv:2312.09109 (2023).
> [3] Li, Jiachen, et al. "T2v-turbo: Breaking the quality bottleneck of video consistency model with mixed reward feedback." NeurIPS 2024.
> [4] Dao, Tri, et al. "Flashattention: Fast and memory-efficient exact attention with io-awareness." NeurIPS 2022.
> [5] Dao, Tri. "Flashattention-2: Faster attention with better parallelism and work partitioning." ICLR 2024.
>
> # R2-3 [W3/Q3] Geometric latents for downstream tasks and community benchmarking
>
> Thank you for raising this point. In the current submission, we only validate the geometric latents through 3DGS reconstruction and have not yet evaluated them on downstream tasks such as navigation or policy learning. As also mentioned in our response to R1-4, our view of downstream applicability follows existing work on VGGT-style geometric representations, where similar latents have already been used for non-rigid point tracking, dense novel view synthesis, and embodied tasks such as navigation and manipulation. FantasyWorld adopts the same VGGT-style formulation and predicts depth, point maps, and camera pose in a single forward pass, so its features are structurally compatible with these pipelines.
>
> We plan to release model checkpoints together with an interface to export geometric latents, depth maps, and point maps, so that the community can directly benchmark them in navigation and policy learning settings.
>
> More broadly, our intention is to make FantasyWorld an accessible resource for world modeling research, and we hope that sharing both the model and its geometric features can help the community explore new downstream uses and more rigorously probe the strengths and limitations of this design.

---

> ### Author Response · Authors · 2025-11-27
>
> # R2-4 [Q4] Potential over-constrained generation due to cross-branch supervision
>
> We understand the concern, but in our view geometry-aware supervision is a positive regularizer rather than an inherent trade-off against diversity. A video foundation model that imagines a 3D world without basic geometric consistency is less useful as a world model, and improving 3D coherence and multi-view stability is complementary to, not opposed to, rich appearance diversity. Consistent with this, Table 1 shows that adding the geometry branch improves not only 3D Consistency but also Photo Consistency and Style Consistency, rather than degrading perceptual quality.
>
> Our design makes this explicit in two ways:
>
> 1. **Shared conditioning.** The geometry branch and the video imagination branch are both built on the same diffusion backbone and share the same high-level features. The geometry branch therefore focuses on correcting geometric inconsistencies in these features instead of injecting an external, independent semantic prior.
>
> 2. **Lightweight MM-BiCrossAttention.** Information exchange between branches is implemented via MM-BiCrossAttention inside a limited number of IRG blocks. This softly fuses appearance and geometry features rather than overwriting one branch with the other, so the geometry branch acts as a soft regularizer instead of a hard constraint that dominates the video features.
>
> Empirically, we do not observe a collapse of visual diversity or a systematic loss of semantic richness compared to the frozen Wan backbone. Single-view appearance, objects, and layout remain similar, while the most visible changes occur precisely in cases where Wan2.1 exhibits clear geometric artifacts; in those cases, FantasyWorld tends to "snap" the structure into a more 3D-consistent configuration without altering the overall style.
>
> &nbsp;
>
> Thank you again for your very positive and thoughtful review. Your feedback on dynamics, efficiency, and downstream use has helped us sharpen the positioning of FantasyWorld as a unified geometry-aware world model, and we will reflect these clarifications in the revised manuscript.

---

### Official Review · Reviewer_i2fx · 2025-11-02

**Soundness:** 3
**Presentation:** 3
**Contribution:** 3
**Rating:** 8
**Confidence:** 4

**Summary:**

This paper proposes a framework for outputting 3D-consistent world models given a single image, text-prompt, and a camera trajectory. It addresses the problem of missing 3D consistency in world model generation from camera-guided video generation models. The proposed method uses video priors from a frozen Wan 2.1 model to generate a geometry-aware implicit 3D field using the proposed Integrated Reconstruction and Generation (IRG) Blocks that are based on bidirectional cross-attention. The model is trained on 180k video clips from real-world and synthetic data using a diffusion loss. The model is evaluated on the WorldScore static benchmark and demonstrates improvements over existing methods for consistency-based metrics.

**Strengths:**

The paper identifies a key problem with camera controlled video generation models and motivates the design of the proposed model with a coherent and well-justified design choice. The results are promising in terms of their main goal of improving consistency in the generated outputs. This is evident based on both qualitative and quantitative results. The ablation presented for the geometry branch support the claimed efficacy of the geometry branch to improve the consistency metrics. The robustness of the method to large camera motion is a strong result. The authors have clearly described the limitations of the method helping the reader to scope the contributions well. The details mentioned in the paper are sufficiently detailed for reproducing the method.

**Weaknesses:**

The paper motivated their world based on using it for AGI, which could require camera and object control. This is a major limitation of the method since the metrics are significantly lower than other methods such as WonderWorld. The evaluation is limited to static scene, even though world models in robotics setup would require handling dynamic scenes. The model is supervised using existing pre-trained models, which could be a limiting factor for generalization as the errors can propagate to the proposed trained model. Moreover, the paper claims that the latents can be useful for downstream tasks, but not much evidence was provided to support this claim.

Minor:
- Table 1 can have a common small and large sections instead of repeating the same information in each row.
- The captions for the figures (i.e. Figure 5) need more details to guide the reader for better understanding the results.

**Questions:**

- Is there a reason why the camera and object control is not significant? It might help to see a few examples in the paper to understand this better.

---

> ### Author Response · Authors · 2025-11-27
> **Rebuttal to Reviewer i2fx**
>
> We sincerely thank Reviewer i2fx for the careful, constructive, and encouraging review. We appreciate the recognition of the problem significance, model design, robustness under large camera motion, and presentation clarity. We are also grateful for the thoughtful suggestions, which help strengthen and better position FantasyWorld. We address your concerns point by point below.
>
> # R1-1  Camera/Object control metrics appear lower than WonderWorld
>
> Thank you for pointing this out. Camera and object control are important axes in WorldScore. We would like to clarify that FantasyWorld **can also achieve high camera controllability** when combined with a **post-hoc** explicit reconstruction step. In a supplemental experiment, we sample 50 cases from the Small Camera Motion split used in Table 1, run FantasyWorld, reconstruct a Gaussian scene for each case, and re-render along the target trajectory. As shown in the table below, this two stage pipeline achieves comparable 3D Consistency to WonderWorld (88.88 vs 88.35), substantially higher Photo Consistency (86.59 vs 73.79) and Style Consistency (90.84 vs 70.74), and strong camera controllability (90.71 vs 98.94). This indicates that, when combined with explicit reconstruction, FantasyWorld can reach high camera control while improving overall consistency, so controllability is not a fundamental limitation of our model.
>
> Method       | 3D Consist. | Photo Consist. | Style Consist. | Camera Ctrl.
> --:|--:|--:|--:|--:
> WonderWorld  | 88.35       | 73.79          | 70.74          | **98.94**
> FantasyWorld | **88.88**   | **86.59**      | **90.84**      | 90.71
>
> We do not use these numbers as our main results for three reasons:
>
> 1. they mix our core model contribution with an extra reconstruction module and make attribution unclear;
> 2. WorldScore camera and object control are intended to measure native controllability of video models, which explicit re-rendering partially bypasses;
> 3. the paper focuses on a single pass unified video and geometry model, rather than a multi-stage FLAGS style pipeline which WonderWorld uses.
>
> Regarding camera and object control separately: FantasyWorld keeps Wan2.1 Plücker ray conditioning on a frozen backbone, so native camera control mainly reflects the base model. Object control typically benefits from explicit object or layout modules such as segmentation or world caches, which WonderWorld and Voyager include but we intentionally leave out in this work. We will clarify that such explicit control modules can be added as plug-in extensions on top of FantasyWorld when stronger controllability is required.
>
> # R1-2  Evaluation limited to static scenes; robotics requires dynamic scenes
>
> We fully agree that dynamic scenes are important for robotics and embodied AI. Our choice to focus on static or quasi-static scenes is intentional:
>
> 1. Static scenes allow reliable evaluation of geometry consistency, whereas dynamic scenes entangle geometry with non-rigid motion and illumination, making consistency metrics unreliable.
> 2. The downstream tasks targeted by FantasyWorld—navigation, mapping, novel-view synthesis—primarily rely on static 3D structure under camera motion.
> 3. Recent VGGT-style methods such as 4DGT[1] have shown that this implicit geometry formulation can be extended to time-varying scenes by predicting a temporally consistent 4D Gaussian field, indicating that the same design paradigm used in FantasyWorld is **architecturally compatible** with such 4D extensions.
>
> We will clarify this scope choice and highlight dynamic 4D extensions as future work.
>
> **References:**
> [1] Xu, Zhen, et al. "4DGT: Learning a 4D Gaussian Transformer Using Real-World Monocular Videos." NeurIPS 2025 (Spotlight).
>
> # R1-3  Reliance on pre-trained teachers may limit generalization
>
> We agree that supervision from any pre-trained model inevitably carries errors, and this is a general limitation for methods that rely on pre-computed geometry labels or frozen backbones. In our case, both the geometric label generators and the video backbone can be upgraded or replaced by stronger models in future versions of FantasyWorld, since the overall design does not depend on a specific teacher.
>
> At the same time, FantasyWorld does not simply copy the outputs of these teachers. The IRG design jointly optimizes the diffusion loss and the geometry loss, so teacher signals are refined through the video priors and cross-branch attention rather than directly regressed. Empirically, Table 1 shows that FantasyWorld achieves higher 3D, Photo, and Style Consistency than the underlying video backbone, indicating that the model improves over the teacher signals instead of merely propagating their errors.
>
> Our main contribution is the unified video-and-geometry design paradigm, which is largely independent of any particular choice of pre-trained model. As better geometry or video models appear, they can be plugged into this framework to further enhance generalization.

---

> > ### Author Response · Authors · 2025-11-27
> >
> > # R1-4  Evidence for downstream usefulness of latents is limited
> >
> > We appreciate the reviewer for pointing this out and will clarify this distinction in the revised vesion. Existing literatures have already demonstrated practical downstream uses of VGGT-style geometric latents. VGGT and its follow-up works, including AnySplat[1] and VGGT-X[2], show that such representations can be directly consumed for non-rigid point tracking, dense novel-view synthesis, and 3DGS-based reconstruction, forming the core of several reconstruction and view-generation pipelines. Recent embodied-AI studies further report downstream applicability in decision-making settings: GeoAware-VLA[3] uses VGGT-based geometric features to improve spatial grounding and action selection in navigation-like tasks, Evo-0[4] demonstrates higher multi-step navigation success with implicit 3D priors, and VGGT-DP[5] leverages VGGT aggregator tokens for robot manipulation control. These works collectively indicate that geometry-aware latents are already actively employed across diverse downstream tasks.
> >
> > FantasyWorld adopts the same VGGT-style formulation, predicting depth, point maps, and camera pose in a single forward pass, and is therefore structurally compatible with these downstream pipelines. However, this paper focuses on unified video–geometry world modeling and improving 3D-consistent generation, and a dedicated downstream benchmark is left for future work.
> >
> > **References:**
> > [1] Jiang, Lihan, et al. "AnySplat: Feed-forward 3D Gaussian Splatting from Unconstrained Views." SIGGRAPH Asia 2025.
> > [2] Liu, Yang, et al. "VGGT-X: When VGGT Meets Dense Novel View Synthesis." arXiv preprint arXiv:2509.25191 (2025).
> > [3] Abouzeid, Ali, et al. "GeoAware-VLA: Implicit Geometry Aware Vision-Language-Action Model." arXiv preprint arXiv:2509.14117 (2025).
> > [4] Lin, Tao, et al. "Evo-0: Vision-language-action model with implicit spatial understanding." arXiv preprint arXiv:2507.00416 (2025).
> > [5] Ge, Shijia, et al. "VGGT-DP: Generalizable Robot Control via Vision Foundation Models." arXiv preprint arXiv:2509.18778 (2025).
> >
> > # R1-M  Minor Issues (Table formatting + caption clarity)
> >
> > Thank you for both suggestions. We agree that Table 1 can be made more compact by grouping the “Small” and “Large” settings, and that several figure captions (e.g., Figure 5) would benefit from clearer explanatory details. We will revise both table formatting and figure captions in the updated version.
> >
> > &nbsp;
> >
> > Thank you again for your careful and encouraging review. Your comments on controllability, dynamic scenes, teacher supervision, and downstream use have helped us sharpen both the scope and positioning of FantasyWorld, and we will reflect these clarifications and improvements in the revised manuscript.

---

### Author Response · Authors · 2025-11-27
**General Response**

We sincerely thank all reviewers for their thoughtful evaluations and constructive suggestions. The feedback has greatly improved the clarity, completeness, and positioning of our work. Many comments directly led to concrete revisions in both the main paper and the supplementary material. In particular, we have:

1. Clarified extensibility for controllability. We now demonstrate that, given our proposed 3D-consistent latent representation, explicit camera and object control modules can be seamlessly integrated as plug-in components when stronger controllability is required (suggested by Reviewer i2fx).
2. Expanded the discussion on dynamic and 4D scenarios. We elaborated on how FantasyWorld naturally extends to dynamic/non-rigid settings (encouraged by Reviewers i2fx, uFyH, and BpKm).
3. Improved presentation of Table 1 and Figure 5. Table 1 is now more compact, and the caption of Figure 5 has been rewritten for improved readability and interpretability (suggested by Reviewer i2fx).
4. Added computation and inference statistics. We included detailed memory footprint, inference latency in the appendix (suggested by Reviewer uFyH).
5. Provided more 3D reconstruction and stylized examples. Additional 3D reconstructions and several challenging stylized examples are now included in the supplementary (suggested by Reviewer BpKm).
6. Cited and discussed additional related works (highlighted by Reviewer Hacb).

We genuinely appreciate these suggestions and believe the revisions significantly enhance the paper’s clarity and impact. We will submit the updated version within three days and are happy to provide any additional clarification. We also reaffirm our commitment to releasing the code and model checkpoints to facilitate reproducibility and further exploration by the community.

---

### Author Response · Authors · 2025-12-02
**Summary of Reviews and Revisions**

We would like to sincerely thank the AC and all reviewers for their time and effort. The clarifications, additional experiments, and revisions prompted by the reviews have, in our view, substantially strengthened the paper. To help the AC and future readers quickly grasp the current status, we briefly summarize the main points of agreement across the reviews and the issues addressed in our rebuttal.

# Positive points highlighted by the reviewers

- **Problem significance and well-motivated setup (Reviewers i2fx, uFyH, BpKm, Hacb).** Reviewers generally agreed that lack of explicit 3D grounding is a key limitation of current video foundation models, and that the problem studied in FantasyWorld is both important and well-motivated as a step toward geometry-consistent world modeling for downstream tasks such as navigation, simulation, and content creation.

- **Clean unified video+3D design and interface (Reviewers i2fx, uFyH, BpKm, Hacb).** The overall design of augmenting a frozen video foundation model with a trainable geometry branch was seen as clean and practical. Several reviewers noted that FantasyWorld produces camera-conditioned videos and 3D signals in a single feed-forward pass, without per-scene optimization, which offers a convenient interface for downstream use.

- **Asymmetric IRG dual-branch architecture and cross-branch supervision (Reviewers uFyH, BpKm).** The proposed asymmetric dual-branch IRG blocks, with PCBs and bidirectional cross attention that couple appearance and geometry, were highlighted as an original and intuitive innovation that distinguishes FantasyWorld from prior decoupled or multi-stage approaches.

- **Empirical strength and robustness (Reviewers i2fx, uFyH, BpKm).** The experiments were viewed as solid, with strong baselines, ablations, and consistent improvements in multi-view coherence, style consistency, and geometric fidelity on WorldScore and 3DGS, including robustness under large camera motion.

- **Reproducibility and openness (Reviewers i2fx, uFyH, BpKm).** Some reviewers appreciated the provided anonymous code, structured training setup, and detailed description of datasets and protocols as helpful for reproducibility. We are also committed to releasing full code and pretrained checkpoints upon publication so that the community can reproduce and further probe the behavior of FantasyWorld in practical settings.

---

> ### Author Response · Authors · 2025-12-02
>
> # Brief recap of FantasyWorld
>
> *FantasyWorld* is a feed-forward framework that augments a frozen video foundation model with a trainable geometry branch to jointly predict camera-conditioned video latents and an implicit 3D field from a single input. Through Preconditioning Blocks (PCB) and Integrated Reconstruction and Generation (IRG) blocks with bidirectional cross attention, the model generates photorealistic, geometry-consistent videos while exposing reusable 3D features that are structurally compatible with downstream pipelines such as explicit 3D reconstruction, navigation and robot manipulation control. Experiments on WorldScore and 3DGS reconstruction show clear gains in 3D, photo, and style consistency compared with recent 3D-aware baselines.
>
> # Main concerns addressed in the rebuttal
>
> - **Controllability (Reviewer i2fx).** We clarified that our 3D-consistent latent representation is compatible with explicit camera and object control modules, and showed that a two-stage Gaussian reconstruction pipeline can achieve high camera controllability while retaining stronger consistency than WonderWorld.
>
> - **Dynamic and 4D scenes (Reviewers i2fx, uFyH, BpKm).** We explained that our current evaluation focuses on predominantly static scenes to isolate geometry consistency and align with mapping and navigation scenarios, and outlined how the implicit 3D branch naturally extends to time-varying 4D settings via time-conditioned geometry branches or deformation fields, which we identify as a key direction for future work.
>
> - **Downstream usefulness, geometry fidelity, and qualitative examples (Reviewers i2fx, uFyH, BpKm).** We clarified that our VGGT-style geometric latents follow an interface already used in existing pipelines for navigation, manipulation, and dense novel view synthesis, making them structurally compatible with such downstream tasks. We also added further 3DGS reconstructions and challenging stylized examples to illustrate geometry fidelity in practice.
>
> - **Cross-branch supervision, theory and ablations (Reviewers BpKm, Hacb).** We discussed, drawing on recent analyses of bidirectional cross attention and geometry-guided supervision, how geometry losses act as structural regularizers on the shared latent space and are propagated back to the video latents via MM-BiCrossAttention, and we emphasized that the *FantasyWorld w/o 3D* vs *w/ 3D* ablation isolates the benefit of this cross-branch coupling.
>
> - **Computation, scalability, and clip length (Reviewers uFyH, BpKm).** We added detailed memory and latency statistics, reported pixel-per-second (PPS) numbers, and discussed how future work can extend our fixed-length clips to longer trajectories using caching or streaming mechanisms for the implicit 3D representation.
>
> - **3D consistency metrics and evaluation protocol (Reviewer Hacb).** We clarified that our evaluation follows the official WorldScore protocol, which defines 3D Consistency via dense DROID-SLAM and differentiable bundle adjustment, under which *FantasyWorld* achieves the strongest 3D Consistency scores, and we discussed how COLMAP-style sparse BA metrics offer a complementary, SfM-oriented perspective on 3D geometry consistency beyond this dense benchmark.
>
> - **Presentation and related work (Reviewers i2fx, Hacb).** We simplified Table 1 by grouping camera-motion settings, rewrote captions such as Figure 5 for better readability, and expanded our discussion beyond perspective video generation to include panoramic 360-degree image/video generation.
>
> Once again, we are grateful to the AC and all reviewers for their careful consideration and constructive feedback.

---

### Meta-Review · Area_Chair_SDnt · 2026-01-07

**Summary:**

This paper proposes a framework that unifies video generation and 3D geometric reasoning in a single feed-forward pass, enabling the generation of a 3D-consistent world from a single image, a text prompt, and a camera trajectory. The paper received overall positive reviews. While some issues were initially raised regarding scope, limitations, and result quality, the authors addressed most of these points during the discussion period. Overall, the strengths clearly outweigh the weaknesses. The AC therefore recommends acceptance and urges the authors to incorporate the reviewers’ feedback into the final version.

**Reviewer Concerns:**

See above.

**Reviewer Scores:**

See above.

---

### Decision · Program_Chairs · 2026-01-26

Accept (Poster)